# Byzantine-Robust Dynamic Weighted Aggregation Framework for Optimal Attack Mitigation in Federated Learning

## Abstract

Federated learning (FL) has emerged as a promising solution to enable distributed learning on sensitive data without centralized storage and sharing. However, FL is vulnerable to data poisoning attacks, where malicious clients aim to manipulate the training process by injecting poisonous data. Existing defense mechanisms for FL suffer from limitations, including a trade-off between precision and robustness, assumptions on asymptotic optimal bounds on error rates of parameters, *i.i.d.* data distributions, and strong-convexity assumptions on the optimization problem. To address these limitations, we propose a novel framework called Federated Learning Optimal Transport (**FLOT**). Our method leverages the Wasserstein barycentric technique to obtain a global model from a set of locally trained models on client devices. Additionally, **FLOT** introduces a loss function-based rejection (LFR) mechanism to suppress malicious updates and a dynamic weighting scheme to optimize the Wasserstein barycentric aggregation function. We evaluate **FLOT** on four benchmark datasets: GTSRB, KBTS, CIFAR10, and EMNIST. Our experimental results demonstrate that **FLOT** outperforms existing baseline methods under single and multi-client attack settings. Also, it serves as a robust client selection technique under no attack. We also prove the Byzantine resilience of **FLOT** to demonstrate its effectiveness. These results underscore the practical significance of **FLOT** as an effective defense mechanism against data poisoning attacks in FL while maintaining high accuracy and scalability. The robustness and effectiveness of **FLOT** make it a promising solution for real-world applications where data privacy and security are critical.

## 1 Introduction

Federated Learning (FL) revolutionizes collaborative machine learning (ML) by establishing a client-server framework that upholds data privacy without necessitating the sharing of sensitive information [Xu et al., 2019a; Guo et al., 2020; Fang et al., 2021; 2020a]. Its practical applications span a wide range, encompassing mobile user personalization Gboard [gbo, 2017], healthcare [Kumar & Singla, 2021], and blockchain [Cao et al., 2023], among others. However, the decentralized nature of FL renders it highly susceptible to adversarial attacks [Mothukuri et al., 2021; Shejwalkar et al., 2022]. Consequently, comprehending the characteristics of such attacks becomes pivotal for ensuring FL security. Hence, this paper focuses on the prevalent and pertinent category of attacks encountered in production deployments, specifically, untargeted black-box online data poisoning attacks as stated in recent research [Shejwalkar et al., 2022]. In this context, attackers aim to induce general misclassifications rather than explicitly targeting particular labels. Nevertheless, **FLOT** can also be applied to defend against white-box poisoning attacks since it is agnostic to the type of attack at the clients.

Existing defenses against data poisoning attacks in FL fall into two primary categories: anomaly detection and innovative model aggregation techniques [Shen et al., 2016; Rieger et al., 2022; Blanchard et al., 2017; Yin et al., 2018]. Anomaly detection methods scrutinize various aspects of client updates to identify malicious clients, while novel aggregation techniques claim to possess Byzantine robustness. However, the latter approach exhibits significant drawbacks, including impractical asymptotic bounds, strong assumptions of *i.i.d.* data distribution, and strongly convex optimization

problems that often do not align with real-world scenarios. To address these limitations and effectively counteract poisoning attacks in FL, we introduce Federated Learning Optimal Transport (**FLOT**), a novel dynamic weighted federated aggregation method founded on Optimal Transport (OT) principles [Monge, 1781], [Kantorovich, 2006].

Our defense strategy is grounded in the premise that updates from a malicious client engaged in data poisoning will exhibit distinguishable characteristics compared to benign client updates, particularly regarding validation loss at the server. This divergence can be identified and addressed through our hypothesis on Loss Function-based Rejection (LFR). Figure 1 provides insight into the validation loss of 10 clients operating under multi-client attack conditions. We observe a clear dispersion in the loss values of malicious clients during the initial rounds, which tend to converge after approximately 60 rounds as the global model is updated with the remaining benign client updates. For the next iteration, all the clients train their local models using the new global model. Previous research [Bhagoji et al., 2019; Fang et al., 2020b] has underscored the efficacy of LFR and accuracy-checking methods for detecting malicious updates in FL. Both of these methods rely on a validation dataset at the server to evaluate the quality of updates received from clients. It is important to note that using a validation dataset

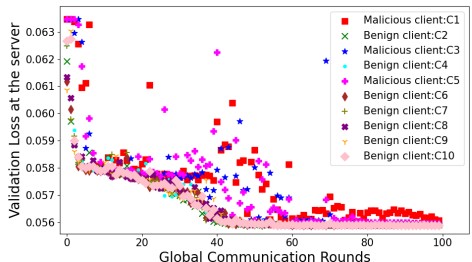

Figure 1: Validation losses of individual client model at the server for 100 global communication rounds under 33% multi-attack settings for the KBTS dataset.

at the server is a well-established practice in the FL field and does not intrude upon clients' privacy. Fang et al. [2020b] have explored methods for implementing a validation dataset without compromising client privacy, such as utilizing a synthetic dataset to mimic the distribution of real data generated by the server [Bhagoji et al., 2019]. **FLOT** aligns seamlessly with existing literature and maintains the versatility of FL applications. Moreover, we harness the advantages of Wasserstein Barycenters [Agueh & Carlier, 2011] for deriving a global model from local models and employ LFR to furnish weighted coefficients for the Wasserstein Barycentric function, thereby facilitating the identification and elimination of malicious updates.

The primary contributions of this work can be summarized as follows: (i) We pioneer the application of OT as an optimization technique to counter data poisoning attacks in the FL domain. To the best of our knowledge, our work represents the first utilization of OT in an adversarial FL context. (ii) We propose **FLOT**, a novel dynamic weighted federated aggregation method and provide a robust solution for securely aggregating gradient updates on a global server. Furthermore, we substantiate the reliability of **FLOT** through theoretical proofs and convergence analyses. (iii) **FLOT** brings about a notable advancement in terms of time complexity. It operates at $\mathcal{O}(nlog(n)d)$ complexity, a substantial improvement compared to the $\mathcal{O}(n^2d)$ complexity associated with the Krum function [Blanchard et al., 2017]. (iv) Our comprehensive evaluation encompasses four widely recognized standard datasets covering diverse FL and attack scenarios. The **FLOT** method consistently delivers superior accuracy and stability under attack conditions across these datasets.

## 2 RELATED WORK

This section reviews the literature in terms of the defenses for FL and OT in ML. Existing attacks in FL are provided in the Appendix. In recent years, several existing defenses have been proposed, including Byzantine robust aggregation methods like Krum [Blanchard et al., 2017], trimmed mean [Yin et al., 2018], median [Yin et al., 2018] in FL. For instance, FLTrust [Cao et al., 2021] enables accurate global model learning even when a bounded number of clients are malicious. However, the performance of FLTrust is highly dependent on the choice of root dataset at the server. LoMar [Li et al., 2023] scores model updates using kernel density estimation in the first phase and determines an optimal threshold to distinguish between malicious and clean updates in the second phase. FL-Defender [Jebreel & Domingo-Ferrer, 2023] analyzes the behaviour of neurons related to the attacks and proposes robust discriminative features using worker-wise angle similarity. Although these methods have shown promising results, they still have limitations, such as the assumption of a

representative root dataset at the server, limited effectiveness in handling complex models, and difficulty in distinguishing malicious from legitimate updates. To overcome these limitations, our proposed method, **FLOT**, utilizes an optimal transport approach and adaptive aggregation weights to limit the impact of malicious updates in FL.

Optimal transport theory is gaining popularity in ML due to its efficiency in various applications [Torres et al., 2021]. It has been used in computer vision for dissimilarity measurement [Rubner et al., 2000] and image-to-image color transfer [Alghamdi et al., 2019; Rabin et al., 2014]. In GANs, OT has been used to improve training stability [Avraham et al., 2019; Salimans et al., 2018; Adler & Lunz, 2018], and WGAN-QC [Liu et al., 2019] uses OT to stabilize the training process. Semantic correspondence across images [Liu et al., 2020], domain adaptation [Courty et al., 2017; Singh & Jaggi, 2020], and graph matching [Xu et al., 2019b] have also benefited from OT. Only a few works have explored the use of OT in FL [Farnia et al., 2022; Wang et al., 2020], but to our knowledge, there is no explicit use of OT in FL to defend against data-poisoning attacks. We propose the first defense mechanism using the OT framework in FL, which shows consistent performance over other state-of-the-art methods across benchmarks.

# 3 PRELIMINARIES

**FL setup.** We consider an FL system that has a server and $n$ clients, where each client $k \in [1, n]$ has its local data indicated as $\mathcal{D}_k$. We ensure a *non-i.i.d.* (non-independent and identically distributed) data distribution by splitting the dataset using Dirichlet distribution [Minka, 2000] by the varying parameter $\beta$ among clients. Further details about Dirichlet distribution and $\beta$ are provided in the Appendix. This client data (commonly referred to as *shard*) is private and cannot be accessed by other clients or the server[]. The objective of FL is to learn global model parameter $\nabla \mathcal{W}_g$ that performs well on the global test data $\mathcal{D}_{test}$. At each round $t$, the central server transmits the current version of the global model (i.e., $\nabla \mathcal{W}_g^t$) to update all $n$ clients. Each client $k$ initializes its local model $\nabla \mathcal{W}^t$ with $\nabla \mathcal{W}_g^t$ and trains it on its local data $\mathcal{D}_k$. After the completion of this local training, the client $k$ calculates the gradient update, i.e., $\nabla \mathcal{W}_k^{t+1} = \nabla \mathcal{W}_k^t - \nabla \mathcal{W}_g^t$. These individual client model updates are returned back to the server, which will be aggregated and used for the next round. In general, synchronous federated weighted averaging (FedAvg) [McMahan et al., 2017] based aggregation is used that is given as

$$\nabla \mathcal{W}_g^{t+1} = \nabla \mathcal{W}_g^t + \sum_{k \in n} \lambda_k \nabla \mathcal{W}_k^{t+1}, \tag{1}$$

where, $\lambda_k = \frac{|\mathcal{D}_k|}{\sum |\mathcal{D}_k|}$, and $\sum_k \lambda_k = 1$. This process continues until the convergence of the global model. Further, as FedAvg is a naive aggregation rule that averages the local model parameters to obtain the global model parameters, it is widely used under non-adversarial settings [Dean et al., 2012; McMahan et al., 2017]. However, FedAvg is not robust under adversarial settings as the attacker can manipulate the global model parameters arbitrarily for this mean aggregation rule when compromising only one client device, as shown in the Definition 3.1 stated by [Blanchard et al., 2017; Yin et al., 2018].

**Definition 3.1** *An aggregation rule $\mathcal{A}$ of the form $\mathcal{A}(\nabla \mathcal{W}_1, \nabla \mathcal{W}_2, \ldots, \nabla \mathcal{W}_n) = \sum_{i=1}^{n} \lambda_i \nabla \mathcal{W}_i$ FedAvg [McMahan et al., 2017], where $\lambda_i > 0$ and $\sum_{i=1}^{n} \lambda_i = 1$, is not byzantine robust as a single malicious client $k$ can prevent convergence by proposing $\nabla \mathcal{W}_k = \frac{1}{\lambda_k} \nabla \hat{\mathcal{W}}_k - \sum_{i=1}^{n-1} \frac{\lambda_i}{\lambda_n} \nabla \mathcal{W}_i$, then $\mathcal{A}(\nabla \mathcal{W}_1, \nabla \mathcal{W}_2, \ldots, \nabla \mathcal{W}_n) = \nabla \hat{\mathcal{W}}_k$, where $\nabla \hat{\mathcal{W}}_k$ is the malicious update from the single byzantine client [Blanchard et al., 2017].*

Hence, we take an optimal transport-based dynamic aggregation approach to improve upon FedAvg and mitigate data poisoning attacks in FL.

**Threat model.** We adopt a threat model that aligns with real-world FL production scenarios, where one or more malicious clients periodically inject poisoned local training data to compromise the local model. Significantly, under this threat model, the malicious clients **cannot interfere with** (a) local training procedure done via trusted execution environments (TEE) [Mondal et al., 2021; Chen et al., 2020], (b) server aggregation algorithm, and (c) communication between client and server. However, **they retain the capability** to (a) access predictions from their local models (in a black-box

manner) for any chosen input data and (b) exert complete control over their local data. As detailed in Section 1, our threat model falls within the scope of **untargeted black-box online data poisoning**, recognized as the most practical and realistic threat in FL, as supported by recent research [Shejwalkar et al., 2022]. *Black-box attack methods.* In this paper, we consider three different black-box online untargeted data poisoning attacks, namely, modified simple black-box attack (MSimBA) [Kumar et al., 2020], data poisoning attack static label flipping (DPA-SLF) [Shejwalkar et al., 2022], and data poisoning attack dynamic label flipping (DPA-DLF) [Shejwalkar et al., 2022] based on their relevance and uptodateness. Further, we found that MSimBA outperforms the other two w.r.t. attack effectiveness. Consequently, we used MSimBA as the target data poisoning attack in all the following experiments. We outline the key dimensions of our threat model, our assumptions regarding the FL setup, and attack methodologies in the Appendix.

**Overview of optimal transport (OT).** Gaspard Monge introduced OT [Monge, 1781], [Kantorovich, 2006] to find the most efficient way to move a unit of mass between two distributions. The aim is to minimize the overall ground cost to move the unit mass from the source distribution to the target distribution. The optimization problem can be given as $\min_{t,\, t \neq \mu_s = \mu_t} \int \mathcal{C}(a, t(a)) \, d\mu_s(a)$, where $\mu_s$, $\mu_t$ correspond to source and target distributions, respectively. $\mathcal{C}(.,.)$ is the ground cost of moving a unit mass between two positions $x, t(x)$. The constraint $t \neq \mu_s = \mu_t$ ensures that the source is completely transported to the target. In general, the OT solution is used in two main aspects: (i) to find the optimal value that measures the similarity between two distributions, also known as Wasserstein distance. (ii) To find the OT matrix, which is the optimal correspondence mapping between distributions. Please refer to the Appendix for details about different OT optimizations. *Wasserstein Barycenters [Agueh & Carlier, 2011]:* It is a distribution that minimizes the weighted sum of Wasserstein distance w.r.t. all other distributions. It aims to find a distribution $\mu$ such that

$$\min_{\mu} \sum_n \alpha_n \mathbb{W}(\mu, \mu_n),  \tag{2}$$

where $\alpha_i$ represent the weight of distribution $\mu_i$, $\mathbb{W}(.,.)$ correspond to Wasserstein distance between distributions given by

$$\mathbb{W}(\mu, \mu_n) = \inf_{\gamma \in \Gamma_{\mu,\mu_n}} \mathbb{E}_{(\mathcal{X},\mathcal{Y} \sim \gamma)} ||\mathcal{X} - \mathcal{Y}||_2^2,  \tag{3}$$

where inf is take over couplings between $\mu$ and $\mu_n$.

**Problem formulation.** Let us assume we are at the $t^{th}$ communication round in FL such that the server receives the model updates from $k$ clients and $\mathcal{D}_v$ is the validation data at the server. Let $\{\nabla \mathcal{W}_1^t, \nabla \mathcal{W}_2^t, \ldots, \nabla \mathcal{W}_k^t\}$ are model updates that correspond to $\{\mathcal{C}_1, \mathcal{C}_2, \ldots, \mathcal{C}_k\}$ clients, respectively. Also, let us assume there are $\rho$ unknown malicious client updates $\rho < n$. Now, the aim is to find a global model weight $\mathcal{W}_g^t$ that minimizes its weighted Wasserstein distance w.r.t. other benign client model weights $\{\nabla \mathcal{W}_1, \nabla \mathcal{W}_2, \ldots, \nabla \mathcal{W}_k\}$ after dynamically discarding the malicious updates.

# 4 OT-BASED APPROACH TO MITIGATE FL POISONING ATTACKS: THEORY

This section presents the theoretical motivation for our OT-based approach to mitigate the problem of data poisoning attacks in FL. Our defense strategy is based on our hypothesis on LFR, such that updates from a malicious client engaged in data poisoning will exhibit distinguishable characteristics compared to benign client updates, particularly in terms of validation loss at the server. Before we explain the proposed defense methodology, we establish the concept of $(\omega, \rho\chi)$ - Byzantine resilience for an aggregation rule as defined in Definition A.1. A more comprehensive proof is available in the Appendix, which elaborates that any aggregation rule rooted in LFR must satisfy Equations 5, 6, and 7. These equations collectively assert that the validation loss, subsequent to discarding malicious updates, highly non-*i.i.d.* updates, or a combination thereof, should consistently exhibit a lower value than the total loss calculated when all client updates are considered.

**Definition 4.1** *Let $\mathbb{N} = \{\nabla \mathcal{W}_1, \nabla \mathcal{W}_2, \ldots, \nabla \mathcal{W}_n\}$ be $n$ total non-i.i.d. set of local clients model updates. Let $\mathbb{R} = \{\nabla \hat{\mathcal{W}}_1, \ldots, \nabla \hat{\mathcal{W}}_\rho\}$ be $\rho$ non-i.i.d. set of Byzantine local clients model updates. Let $\mathbb{X} = \{\nabla \hat{\mathcal{W}}'_1, \ldots, \nabla \hat{\mathcal{W}}'_\chi\}$ be $\chi$ highly non-i.i.d. set of benign local clients model updates. An aggregation rule $\mathcal{A}$ is said to be $(\omega, \rho\chi)$-Byzantine Resilient) if for any $1 \leq \cdots \leq i_1 \cdots \leq i_\rho \cdots \leq j_1 \leq \cdots \leq j_\chi \leq \ldots n$, vector*

$$\mathcal{A}(\nabla \mathcal{W}_1, \ldots, \nabla \hat{\mathcal{W}}_1, \ldots, \nabla \hat{\mathcal{W}}_\rho, \ldots, \nabla \hat{\mathcal{W}}'_1, \ldots, \nabla \hat{\mathcal{W}}'_\chi, \ldots, \nabla \mathcal{W}_n)  \tag{4}$$

*satisfies the following*

$$\sum_{\nabla \mathcal{W}_k \in (\mathbb{N} \setminus \mathbb{R})} \mathcal{L}(\mathcal{D}_v, \nabla \mathcal{W}_k) \leq \sum_{\nabla \mathcal{W}_k \in \mathbb{N}} \mathcal{L}(\mathcal{D}_v, \nabla \mathcal{W}_k), \tag{5}$$

$$\sum_{\nabla \mathcal{W}_k \in (\mathbb{N} \setminus \mathbb{X})} \mathcal{L}(\mathcal{D}_v, \nabla \mathcal{W}_k) \leq \sum_{\nabla \mathcal{W}_k \in \mathbb{N}} \mathcal{L}(\mathcal{D}_v, \nabla \mathcal{W}_k), \tag{6}$$

$$\left\| \sum_{\nabla \mathcal{W}_k \in \mathbb{N}} \mathcal{L}(\mathcal{D}_v, \nabla \mathcal{W}_k) - \sum_{\nabla \mathcal{W}_k \in \mathbb{N} \setminus (\mathbb{R} \cup \mathbb{X})} \mathcal{L}(\mathcal{D}_v, \nabla \mathcal{W}_k) \right\| \geq \omega, \tag{7}$$

*for some $\omega \geq 0$. Here, $\mathcal{L}(\mathcal{D}_v, \nabla \mathcal{W}_k)$ denote the loss of $\nabla \mathcal{W}_k$ model on validation data $\mathcal{D}_v$.*

## 5 **FLOT**: METHODOLOGY

In this section, we introduce **FLOT**, an OT-based $(\omega, \rho\chi)$ - Byzantine resilient dynamic weighted federated aggregation rule to mitigate poisoning attacks as defined in Section 4.

Blanchard et al. [2017] prove that *no linear combination* of the vectors can tolerate a single Byzantine worker (Definition 3.1). Specifically, FedAvg [McMahan et al., 2017] is not Byzantine resilient. Existing Byzantine robust algorithms like Krum [Blanchard et al., 2017] select the local model updates representative of most client models by computing the pairwise distances between individual models. However, when the data across the workers are highly non-*i.i.d.*, there is no 'representative' client model. The local client models show high variance with respect to each other as they compute their local gradient over vastly diverse local data. Hence, for convergence, it is crucial to not only select a good (non-Byzantine) local model but also ensure that each of the good models is selected with roughly equal frequency. Further, when applied to non-*i.i.d.* datasets, Krum performs

Figure 2: Overview of **FLOT** integrated FL system with $n$ clients $(C_1, C_2, \ldots, C_n)$. The malicious client $(C_2)$ sends malicious update $(\nabla \mathcal{W}_2^t)$ using poisoning the training data. The central server receives the gradients and performs **FLOT** to obtain the global model $\nabla \mathcal{W}_g^t$.

poorly even without any attack [He et al., 2020]. This is because Krum primarily selects models from $n - c - 2$ (where $c$ is the number of malicious clients), local models whose pairwise distances are closer to others. Hence, the robust aggregation rules may fail on realistic non-*i.i.d.* datasets.

To address this issue, we consider LFR with OT optimization to develop a Wasserstein barycentric aggregation rule called **FLOT**, as shown in Figure 2. In the end, through our experimental results, we show that our **FLOT** also serves as a robust client selection technique in discarding the benign clients that do not perform well on the validation data. This implies that dropping some less performing benign updates helps to improve the accuracy, which also supports the claims of the recent work, DivFL [Balakrishnan et al., 2021].

Now, we explain our **FLOT** framework, as shown in Algorithm 1. To start with, we find the optimal coefficient set of the client model

---

**Algorithm 1** Federated Learning Optimal Transport (**FLOT**) method

---

**Input:** $\nabla \mathcal{W}_n^t$, $n$ client updates for $t^{th}$ round; $\mathcal{D}_v$, validation data at the server
**Output:** $\nabla \mathcal{W}_g^{t+1}$, updated global model
$\alpha = \{\}$     ▷ *LFR based weight multiplier vector*
**for** $i = 1$ to $n$ **do**     ▷ *Loop through n models*
    $\alpha \leftarrow \mathcal{L}(\mathcal{D}_v, \nabla \mathcal{W}_n^t)$     ▷ *Validation loss*
$\alpha' \leftarrow |\alpha - \max(\alpha)|$
$\alpha' \leftarrow$ normalize$(\alpha')$    ▷ *s.t.* $\alpha_i' \in [0,1], \forall i \in n$
$\mathcal{M} \leftarrow$ **FLOT** cost matrix
$\nabla \mathcal{W}_n^t \leftarrow$ ot.lp.barycenter$(\nabla \mathcal{W}_n^t, \mathcal{M}, \alpha')$ ▷ *FLOT aggregator*
return $\nabla \mathcal{W}_n^t$

---

weights $\alpha$ based on loss on validation data $\mathcal{D}_v$, i.e., $\mathcal{L}_v$ of every client model $\nabla \mathcal{W}_i$. It can be formulated as $\alpha \leftarrow \mathcal{L}_v(\mathcal{D}_v, \nabla \mathcal{W})$, $\alpha' \leftarrow |\alpha - \max(\alpha)|$. Now, we define a set $\alpha_0' = \alpha'$ and write $\beta_1 := \{b \in \alpha_0' : b \leq a \, \forall a \in \alpha_0'\}$. Next, we define $\alpha_1' := \alpha_0' \setminus \beta_1$ which discards the highly malicious weight coefficient from the set $\alpha_0'$. Further, we inductively write

$\beta_k := \{b \in \alpha'_{k-1} : b \le a \,\forall\, a \in \alpha'_{k-1}\}, \alpha'_k := \alpha'_{k-1} \setminus \beta_k$, such that $\alpha'_k$ is the final set after discarding $k$ malicious client updates whose $\alpha' = 0^1$. Further, we normalize $\alpha'_k$ to $[0, 1]$ through the softmax of all weighting factors, which is defined as $\alpha'_k = \frac{e^{\alpha'_k}}{\sum_{k=1}^n e^{\alpha'_k}}$.

Now, our optimization problem can be formulated in terms of Wasserstein barycenter as per Eq. 2 as

$$\textbf{FLOT}(\nabla\mathcal{W}_1^t, \nabla\mathcal{W}_2^t, \ldots, \nabla\mathcal{W}_n^t) \leftarrow \min_{\nabla\mathcal{W}_g^t} \sum_k \alpha'_k \mathbb{W}(\nabla\mathcal{W}_g^t, \nabla\mathcal{W}_k), \tag{8}$$

where $t$ is the global communication round.

**Lemma 5.1** *The expected time complexity of our **FLOT**$(\nabla\mathcal{W}_1^t, \nabla\mathcal{W}_2^t, \ldots, \nabla\mathcal{W}_n^t)$ function is $\mathcal{O}(nlog(n)d)$, where, $\nabla\mathcal{W}_1^t, \nabla\mathcal{W}_2^t, \ldots, \nabla\mathcal{W}_n^t$ are d-dimensional vectors.*

*Proof.* Firstly, the parameter server computes the maximum of loss values $(\alpha_1, \alpha_2, \ldots, \alpha_n)$ and updates all its elements $|\alpha - max(\alpha)|$ in $\mathcal{O}(nd)$ time. Then the server selects the loss that is less than a certain threshold (expected time $\mathcal{O}(nlog(n)d)$ with a binary search). Next, it computes the set difference to discard the highly malicious weight vector in $\mathcal{O}(nd)$ time. Finally, the server normalizes the remaining $n - k$ values in $\mathcal{O}(nd)$ time. Hence, adding all the times, we obtain the overall time complexity of **FLOT** as $\mathcal{O}(nlog(n)d)$.

We report that **our proposed FLOT time complexity is $\mathcal{O}(nlog(n)d)$ which is a significant improvement over $\mathcal{O}(n^2d)$ of the Krum function [Blanchard et al., 2017].**

It is important to note that **FLOT** is designed to be highly efficient by only considering the impact of a small subset of clients on the global model rather than all clients. This is achieved through LFR, where only the clients with the smallest loss impact on the global model are considered for further processing. This significantly reduces the number of clients that need to be considered, reducing the computational cost. In practice, **FLOT** can be further improved by using parallel computations at the server along with model compression and quantization techniques.

## 6 FLOT: CONVERGENCE ANALYSIS

In this section, we analyze the convergence of **FLOT** global model aggregation for convex problems under non-*i.i.d.* data setting. Our **FLOT** optimization function, as per Eq. (8), is given by

$$\textbf{FLOT}(\nabla\mathcal{W}_1, \nabla\mathcal{W}_2, \ldots, \nabla\mathcal{W}_n) \leftarrow \min_{\nabla\mathcal{W}_g} \sum_k \alpha'_k \mathbb{W}(\nabla\mathcal{W}_g, \nabla\mathcal{W}_k). \tag{9}$$

Rewriting it, we get the **FLOT** Barycenter functional as

$$\nabla\mathcal{W}_g^* \in \arg\min_{\nabla\mathcal{W} \in \mathcal{P}_2(\mathbb{R}^d)} \alpha'_k \sum_{i=1}^k \mathbb{W}_2^2(\nabla\mathcal{W}_g, \nabla\mathcal{W}_k) =: 2FLOT(\nabla\mathcal{W}_g)^2, \tag{10}$$

(*from Wasserstein-2 spaces ($\mathbb{W}_2^2$)- it is the metric space of probability measures $\mathcal{P}_2(\mathbb{R}^d)$, equipped with the Wasserstein distance as given in Eq. (3)*). The aim is to minimize **FLOT**$(\nabla\mathcal{W}_g)$. Further, we can write the Wasserstein gradient of the above formulation using the Brenier map [Ambrosio et al., 2005] as

$$\nabla\textbf{FLOT}(\nabla\mathcal{W}_g) = -\alpha'_k \sum_{i=1}^k (\mathcal{T}_{\nabla\mathcal{W}_g \to w_i} - \tau), \tag{11}$$

---

[1]Since all the local models are trained on different amounts of non-*i.i.d.* data, all $\alpha'_i s$ are different, where $i \in [1, n]$.

[2]We scaled to one half so that when the derivate is taken the term 2 goes away.

where $\mathcal{T}_{\nabla\mathcal{W}_g \to \nabla\mathcal{W}_i}$ is the Brenier map, $\tau$ is the identity that gives the displacement map of $\nabla\mathcal{W}_g$. Finally, the gradient descent of the global model over $\mathbb{W}$ metric space is given by

$$
\begin{aligned}
\nabla\mathcal{W}_g^{t+1} &= (\tau - \eta_t \nabla\mathbf{FLOT}(\nabla\mathcal{W}_g))_{\#}\nabla\mathcal{W}_g^t \\
&\implies \nabla\mathcal{W}_g^t - (\tau - \eta_t \nabla\mathbf{FLOT}(\nabla\mathcal{W}_g)) \\
&= (\tau + \alpha_k' \sum_{i=1}^{k}(\mathcal{T}_{\nabla\mathcal{W}_g \to w_i} - \tau)_{\#}\nabla\mathcal{W}_g)^t; (Eq.(11)) \\
&= (1 - \eta_t)\nabla\mathcal{W}_g^t + \eta_t \alpha_k' \sum_{i=1}^{k}\mathcal{T}_{\nabla\mathcal{W}_g \to \nabla\mathcal{W}_i}(\nabla\mathcal{W}_g)^t.
\end{aligned}
\tag{12}
$$

Further, we apply the Polyak-Łojasiewicz (PL) inequality [Karimi et al., 2016] given by

$$
f(x) - \inf f \le C||\nabla f(x)||^2, \forall x,
\tag{13}
$$

followed by smoothness of gradient [Mai & Johansson, 2020] given by

$$
f(y) - f(x) \le \langle \nabla f(x), y - x \rangle + \frac{\beta}{2}||y - x||^2,
\tag{14}
$$

for some function $f(x)$, the derivative of $f$ as $\nabla f(x)$ and constant $C$, to prove the linear rate (exponentially) of convergence for gradient descent. Finally, the linear rate of convergence of **FLOT** for gradient descent is given by

$$
\mathbf{FLOT}(\nabla\mathcal{W}_g^{t+1}) - \mathbf{FLOT}(\nabla\mathcal{W}_g^t) \lesssim e^{-\frac{\alpha_k'}{2C}t}.
\tag{15}
$$

# 7 EXPERIMENTS

**Datasets and implementation details.** We extensively evaluate our **FLOT** method using four benchmark datasets for image classification: GTSRB [Stallkamp et al., 2011], KBTS [Mathias et al., 2013], CIFAR10 [Cohen et al., 2017], and EMNIST [Cohen et al., 2017]. We configured FL with a total number of clients as 30, 10, 30, and 10,000 for GTSRB, KBTS, CIFAR10, and EMNIST datasets, respectively. Further, we partition the dataset as 70% for training, 10% for validation at the server, and 20% for testing. Adequate samples were reserved in the validation dataset (10%) to distinguish between malicious and benign updates before aggregation using **FLOT** for global model generation. Our evaluation encompassed two attacker settings: *single-client* and *multi-client*. For multi-client attacks, we introduced varying percentages of adversaries 33%, 50%, specifically 10, 15 randomly selected malicious clients for GTSRB and CIFAR-10 evaluations, and 3, 5 for KBTS. For EMNIST, we explored scalability by considering five different attack percentages 10%, 20%, 30%, 40%, 50%. Also, for the EMNIST dataset, the server randomly selects 100 clients from a pool of 10,000, designating 10, 20, 30, 40, 50 as malicious based on the attack percentages. Each experiment was conducted thrice, and results were averaged with standard deviations presented.

We designed a custom 4-layer CNN architecture followed by two fully connected layers, considering it as the global model for the GTSRB, KBTS, and CIFAR-10 datasets. Furthermore, for a comprehensive evaluation of **FLOT** across various model architectures, we employed ResNet18 [He et al., 2015] for the CIFAR-10 dataset and LeNet5 [LeCun et al., 1998] for EMNIST. We employed the black-box and active data poisoning technique for our default evaluation attack, MSimBA [Kumar et al., 2020]. Furthermore, we conducted evaluations using two recently developed state-of-the-art label-flip attacks in the FL domain: DPA-SLF [Shejwalkar et al., 2022] and DPA-DLF [Shejwalkar et al., 2022]. For more detailed information on the datasets, CNN architectures, data splits, distribution, specific FL parameters, and attack methods, please refer to the Appendix.

**Baselines and evaluation metrics.** We have selected the following state-of-the-art defense baseline techniques based on their up-to-dateness and relevance. Then, we categorized them into four categories for better evaluation: (i) **ND (no defense):** This category includes the FedAvg method [McMahan et al., 2017]. (ii) **CS (client selection):** Within this category, we have considered techniques such as random sampling (RS), Power-of-choice (PC) [Cho et al., 2020], and DivFL balakrishnan2021diverse. (iii) **BzA (Byzantine aggregation):** This group encompasses aggregation techniques designed for byzantine robustness, such as Krum [Blanchard et al., 2017], Trimmed Mean

(TM) [Yin et al., 2018], and Median [Yin et al., 2018]. **(iv) RD (recent defense):** In this category, we have included the very recent defense methods FLTrust [Cao et al., 2021], LoMar [Li et al., 2023], and FLDefender [Jebreel & Domingo-Ferrer, 2023]. This categorization provides a comprehensive framework for evaluating **FLOT** against the current state-of-the-art techniques in the field. We use the maximum classification global test accuracy ($GTA \in [0, 100]\%$) for all global epochs as an evaluation metric. More details are in the Appendix.

**Results discussion.** We conducted baseline evaluations without any attacks or defenses to establish the accuracy of our FL configuration. The results, summarized in Table 1, revealed GTA values ranging from 88.34% to 91.23% across datasets. Notably, the EMNIST dataset exhibited slightly lower performance, likely due to its unique characteristics involving non-*i.i.d.* data distribution among a large pool of 10,000 clients, with aggregation from a random subset of 100 clients. Table 2

Table 1: No attack no defense global test accuracy GTA% ($\uparrow$) performance comparison.

| Dataset | GTA (%) |
|---|---|
| GTSRB [Stallkamp et al., 2011] | $89.80_{\pm 0.41}$ |
| KBTS [Mathias et al., 2013] | $90.02_{\pm 1.16}$ |
| CIFAR10 [Krizhevsky et al., 2009] | $91.23_{\pm 0.27}$ |
| EMNIST [Cohen et al., 2017] | $88.34_{\pm 0.21}$ |

presents the performance of our **FLOT** framework compared to baselines on four benchmark datasets under single-client (1A) and multi-client (50%) attack settings for brevity. Our **FLOT** consistently outperforms other methods across all datasets and attack scenarios.

*FLOT variation (FLOT+RS).* We also evaluated the performance of **FLOT** with random sampling (RS) and observed improvements. **FLOT+RS** achieved approximately 0.8% to 3% higher performance than **FLOT** for the GTSRB and EMNIST datasets. In single-client attack scenarios, where the number of benign clients is one less than the total, all baselines, including Byzantine aggregation techniques, performed similarly to mitigate the impact of a single malicious client. Conversely, **FLOT** exhibited superior performance in multi-client attack settings, with improvements of approximately 1% to 10%. Power-of-choice and DivFL, effective client selection techniques in clean data settings, performed poorly under attack conditions. The non-*i.i.d.* data distribution among clients and strong data poisoning attacks led to reduced performance of Krum, which relies on strong *i.i.d.* assumptions. Additionally, for GTSRB and EMNIST datasets with 30 and 100 selected clients, respectively, **FLOT+RS** outperformed **FLOT**, benefiting from the availability of a large number of clients. However, applying RS to the KBTS dataset with only ten clients resulted in a performance drop when combined with **FLOT**, particularly under higher attack percentages. In the EMNIST dataset setup, where the server randomly selects 100 clients for aggregation, the performance of FedAvg and RS is the same, as shown in Table 2.

*Evaluation on non-i.i.d. data.* To assess our **FLOT**'s robustness in addressing highly non-*i.i.d.* scenarios, we conducted experiments on the CIFAR10 dataset, varying data distribution by adjusting $\beta$ values (0.1, 0.5, 1, 5, and 10). Lower $\beta$ values led to sparse and unbalanced data among clients, occasionally resulting in some clients lacking data for specific classes. Conversely, higher $\beta$ values created densely balanced data distributions with more samples per class assigned to each client. For consistency, we selected $\beta$=1 as the default for all our experiments. We evaluated our method in scenarios with no attack, single-client attack, and multi-client attack with 50% malicious clients on CIFAR10, focusing on brevity. To ensure fairness, we compared our method to existing techniques, including FedAvg, Krum, DivFL, LoMar, and FLDefender, representing the best performers in their respective defense categories. Summarized results are presented in Figure 3. Under the no attack setting, our **FLOT** approach outperformed the FL baseline by more than 1% for CIFAR10, with similar results observed for other datasets. This demonstrates that under no attack conditions, **FLOT** effectively serves as a robust client selection method, prioritizing client updates that enhance overall accuracy. Our findings highlight **FLOT**'s superior performance, particularly in scenarios with diverse updates, including poisoned and highly non-*i.i.d.* updates. In single-client attack conditions, DivFL and Krum perform poorly as they are tailored for well-behaved and *i.i.d.* updates, respectively. Under 50% maliciousness, DivFL performs inadequately, followed by FedAvg without any defense and Krum. Additionally, as non-*i.i.d.* degrees decrease ($\beta$ increases), all evaluated methods exhibit improved performance. Please refer to the Appendix for additional experimental results and an ablation study covering other attacks and settings.

*In summary, our Wasserstein barycenter-based optimization, combined with dynamically weighted coefficients, effectively interpolates between multiple client updates [Lacombe et al., 2022]. This process helps to warp the updates, suppressing malicious ones and enhancing overall performance. FLOT configurations consistently outperformed all baselines under various attack scenarios and maintained a close performance to the FL baseline, with differences exceeding 1% in a no-attack*

Table 2: Comparison of GTA% (↑) with FedAvg no defense (ND), existing client selection (CS) methods, Byzantine aggregation (BzA) rules, and recent robust FL defense (RD) methods. We present single-client attack and multi-client (50%) MSimBA attack results for brevity (please refer Appendix for results of other multi-client attack settings). ==Result==, [result] indicates the best and second best result, respectively, for each attack setting.

| Defense Method | Type | GTSRB 1A | GTSRB 50% | KBTS 1A | KBTS 50% | CIFAR10 1A | CIFAR10 50% | EMNIST 1A | EMNIST 50% |
|---|---|---|---|---|---|---|---|---|---|
| FedAvg [McMahan et al., 2017] | ND | $83.24_{\pm0.80}$ | $30.31_{\pm1.82}$ | $83.26_{\pm1.25}$ | $33.86_{\pm0.53}$ | $85.03_{\pm0.60}$ | $23.53_{\pm0.55}$ | $83.19_{\pm0.45}$ | $20.95_{\pm1.19}$ |
| RS [McMahan et al., 2017] | | $84.45_{\pm0.56}$ | $35.12_{\pm1.02}$ | $84.24_{\pm0.81}$ | $40.15_{\pm0.98}$ | $82.98_{\pm1.02}$ | $25.86_{\pm1.74}$ | $83.19_{\pm0.45}$ | $20.95_{\pm1.19}$ |
| PC [Cho et al., 2020] | CS | $81.29_{\pm0.93}$ | $31.86_{\pm0.15}$ | $80.27_{\pm0.65}$ | $43.93_{\pm1.75}$ | $73.86_{\pm0.28}$ | $22.15_{\pm0.60}$ | $83.15_{\pm1.79}$ | $22.83_{\pm0.44}$ |
| DivFL [Balakrishnan et al., 2021] | | $82.63_{\pm0.13}$ | $32.42_{\pm0.79}$ | $81.63_{\pm1.94}$ | $41.74_{\pm1.63}$ | $74.12_{\pm1.81}$ | $21.52_{\pm0.61}$ | $84.12_{\pm0.66}$ | $24.15_{\pm1.51}$ |
| Krum [Blanchard et al., 2017] | | $\underline{85.80}_{\pm0.59}$ | $39.87_{\pm0.88}$ | $84.29_{\pm1.08}$ | $47.65_{\pm1.98}$ | $85.12_{\pm1.59}$ | $35.48_{\pm1.38}$ | $85.45_{\pm0.49}$ | $25.42_{\pm1.08}$ |
| TM [Yin et al., 2018] | BzA | $82.87_{\pm1.56}$ | $38.15_{\pm1.54}$ | $84.09_{\pm0.85}$ | $44.86_{\pm1.01}$ | $84.43_{\pm1.23}$ | $30.68_{\pm1.61}$ | $84.98_{\pm1.71}$ | $19.36_{\pm0.45}$ |
| Median [Yin et al., 2018] | | $83.39_{\pm0.72}$ | $38.74_{\pm1.38}$ | $84.97_{\pm0.21}$ | $45.28_{\pm0.37}$ | $83.36_{\pm0.18}$ | $33.92_{\pm1.55}$ | $84.45_{\pm0.65}$ | $17.24_{\pm1.74}$ |
| FLTrust [Cao et al., 2021] | | $10.32_{\pm1.89}$ | $7.95_{\pm0.59}$ | $9.49_{\pm0.70}$ | $5.42_{\pm0.32}$ | $8.45_{\pm1.10}$ | $6.96_{\pm1.89}$ | $5.60_{\pm1.16}$ | $4.32_{\pm0.35}$ |
| LoMar [Li et al., 2023] | RD | $84.67_{\pm0.91}$ | $45.98_{\pm0.64}$ | $84.68_{\pm1.72}$ | $58.45_{\pm0.93}$ | $\underline{85.48}_{\pm0.81}$ | $61.76_{\pm0.33}$ | $85.62_{\pm1.85}$ | $53.28_{\pm0.26}$ |
| FLDefender [Jebreel & Domingo-Ferrer, 2023] | | $85.29_{\pm1.50}$ | $49.36_{\pm0.21}$ | $84.74_{\pm0.57}$ | $62.37_{\pm1.67}$ | $84.92_{\pm1.98}$ | $\underline{63.48}_{\pm0.87}$ | $85.73_{\pm1.12}$ | $49.68_{\pm0.32}$ |
| **FLOT** | ours | $85.12_{\pm0.58}$ | $\underline{61.23}_{\pm0.36}$ | $\mathbf{85.94}_{\pm0.46}$ | $\mathbf{69.23}_{\pm0.30}$ | $85.21_{\pm0.64}$ | $\mathbf{64.34}_{\pm1.73}$ | $\mathbf{86.12}_{\pm1.53}$ | $\underline{52.42}_{\pm0.98}$ |
| **FLOT+RS** | | $\mathbf{85.98}_{\pm1.06}$ | $\mathbf{63.45}_{\pm1.45}$ | $\underline{85.02}_{\pm0.72}$ | $\underline{67.46}_{\pm0.15}$ | $\mathbf{86.24}_{\pm1.21}$ | $62.87_{\pm0.69}$ | $\mathbf{86.48}_{\pm0.41}$ | $\mathbf{55.26}_{\pm0.87}$ |

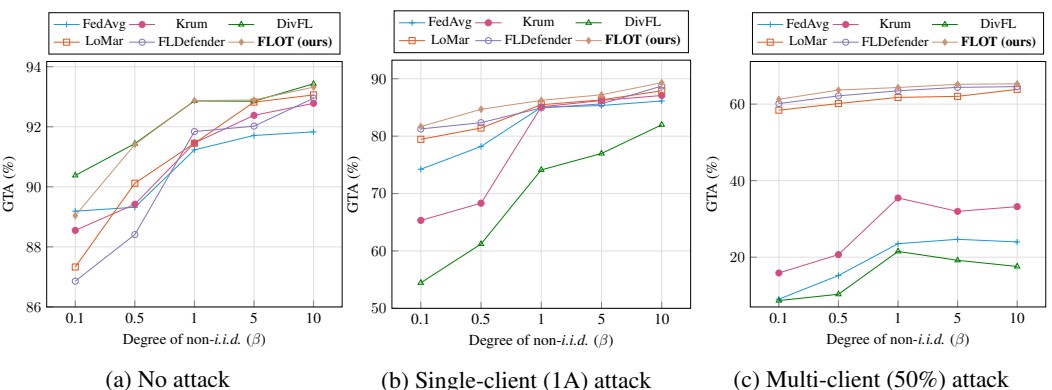

Figure 3: Comparison of GTA% (↑) for different defense techniques under non-*i.i.d.* data distribution (Dirichlet $\beta$) scenarios on CIFAR10 dataset, with MSimBA no attack, single-client attack, and multi-client (50%) attack settings.

*scenario. Additionally, **FLOT** outperformed existing techniques by more than 0.5% and 10% in single-client and multi-client attack settings, respectively, highlighting its Byzantine robustness in the face of non-i.i.d. data poisoning attacks.*

# 8 CONCLUSION

This paper introduces **FLOT**, an optimal transport-based dynamic weighted federated aggregation method designed to mitigate untargeted data poisoning attacks within the FL framework. **FLOT** effectively interpolates global model updates by employing loss-based weighted coefficients and leverages OT optimization via Wasserstein barycenters to obtain a smoothed global model while discarding malicious updates. Our extensive experimental results demonstrate that **FLOT** configurations consistently outperform other methods, including Byzantine robust aggregation rules, in terms of classification performance under both single-client and 50% Byzantine worker scenarios. Additionally, our time complexity analysis reveals a logarithmic improvement ($log(n)$) over the Krum aggregation rule, with the number of clients denoted as $n$. We have also established the $(\omega, \rho\chi)$ - Byzantine resilience of **FLOT**, along with its convergence properties. In the future, we plan to explore various OT optimization variations, including regularization methods to address higher levels of non-*i.i.d.*ness and extend the applicability of **FLOT** to other computer vision tasks such as object detection and segmentation.

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

# A  APPENDIX

In this section, we present additional information that was not included in the main paper due to space limitations. We have meticulously organized the details into individual sections to enhance clarity and facilitate a comprehensive understanding of our work.

## A.1  ADDITIONAL DISCUSSION OF RELATED WORK

This section presents a broader related work regarding existing poisoning attacks in FL. Adversarial attacks in FL can be categorized into data poisoning or model poisoning attacks. In both cases, the attack can be targeted (i.e., to have a specific misclassification) or untargeted (i.e., to induce any misclassification).

**Data Poisoning Attacks:** Adversarial attacks against ML models and deep neural networks have received much attention [Goodfellow et al., 2014; Carlini & Wagner, 2017]. These attacks have been studied mainly for *centralized ML* [Szegedy et al., 2013; Shafahi et al., 2018; Li et al., 2020], without much prior work on untargeted black-box data poisoning attacks on FL settings. Bagdasaryan et al. [2020] proposed a backdoor FL attack framework that trains on the backdoor data using our constrain-and-scale technique and submits the resulting corrupted model as an update to the central server. Fang et al. [2020b] formulated labelflip attacks as optimization problems and applied them to Byzantine-robust federated learning methods. Shejwalkar et al. [2022] proposed two different data poisoning (DP) attacks, namely static labelflip (DP-SLF) and dynamic labelflip (DP-DLF) in

FL. Each of these attack methods serves a unique purpose in highlighting the vulnerabilities and risks associated with FL systems. Bagdasaryan et al. [2020] approach sheds light on the potential for backdoor attacks and emphasizes the need for robust defenses against such threats. Zhang et al. [2020] showcases the effectiveness of generative adversarial attacks in poisoning FL systems. Fang et al. [2020b]'s formulation of labelflip attacks contributes to the development of Byzantine-robust FL techniques by exposing the susceptibility of the learning process to label manipulation. Finally, Shejwalkar et al. [2022]'s DP attacks provide insights into the risks posed by poisoning the training data in FL, highlighting the need for effective detection and mitigation strategies.

**Model Poisoning Attacks:** In this second category, the attacker directly sends malicious updates [Bhagoji et al., 2019; 2018]. Research has been done on ways to create malicious updates effectively. Baruch et al. [2019] proposed a little is enough (LIE) attack by adding noise to the average of the benign updates using the standard deviation of available benign updates to compute a poisoned update. Shejwalkar & Houmansadr [2021] produces malicious model updates by maximally perturbing the benign reference aggregate in the malicious direction. Fang et al. [2020b] compute the average of the benign updates, determine a static malicious direction, and then calculate a poisoned update by finding a suboptimal parameter that circumvents the target aggregation rule.

Each of these attack methods illustrates the vulnerabilities and risks associated with malicious updates in federated learning. Baruch *et al.*'s LIE attack emphasizes the potential impact of injecting noise into the aggregation process, even in small quantities. Shejwalkar *et al.*'s approach showcases the ability to manipulate the learning process by perturbing the benign reference aggregate. Fang *et al.*'s method demonstrates how the strategic selection of updates can undermine the aggregation rule and compromise the quality of the federated model.

*In summary, these attack methods collectively demonstrate various aspects of FL vulnerability, including backdoors, poisoning attacks, label manipulation, and malicious updates. Understanding and addressing these different attack vectors is crucial for enhancing the security and trustworthiness of FL systems.*

In this paper, we focus on defending against **untargeted black-box data poisoning attacks** in FL, as it is the most common and relevant type of attack in production deployments as stated in [Shejwalkar et al., 2022]. These attacks can affect a large population of FL clients and remain undetected for an extended period. **Nonetheless, FLOT can also be applied to defend against white-box poisoning attacks since it is agnostic to the type of attack at the clients.**

## A.2   MORE DETAILS ABOUT OUR THREAT MODEL

In this section, we present the critical dimensions of our threat model and the assumptions we make about the FL setup, as shown in Table 3.

Table 3: Key dimensions of our threat model and their attributes.

| Objective | | | Knowledge & Capabilities | | Attack Mode |
|---|---|---|---|---|---|
| **Security violation** | **Attack specificity** | **Error specificity** | **Model** | **Data distribution** | **Consciously active** |
| **Availability:** Misclassify test data and cause disruption to benign clients' objectives. | **Indiscriminate:** Misclassify all or most of the test inputs during inference. | **Untargeted:** Misclassify the give test data to any other class. | **Black-box:** Adversary cannot break into the compromised clients and cannot manipulate the model parameters. | The adversary can only access the local data distributed at the clients. | **Online:** The adversary repeatedly and adaptively poisons the model based on the attack strategy. |

**Attacker objectives:** The main goal of the attacker is to make the global model (i.e., the one used to perform testing on the server) misclassify all or most of the test data and thereby reduce the performance. The attacker is interested in generic misclassification (untargeted) rather than specific misclassification (targeted).

**Attacker knowledge & Capabilities:** We assume the attacker has the following capabilities on the server and compromised clients.

**Server side.** We assume that the server is a black-box to the attacker. As such, the attacker has *no access to parameters, predictions of the global model, or the aggregation algorithm at the server*. Also, the server is trustworthy and incurious about the model updates. We consider this setup based on recent stated work [Shejwalkar et al., 2022].

**Client side.** We assume that the attacker controls the data used in one (single-client attack) or more clients (multi-client attack). The clients use this data to compute their updates via trusted local model training [Chen et al., 2020; Mondal et al., 2021]. The attacker cannot break into compromised clients' training procedures. Precisely, the attacker can only manipulate the local data of the compromised clients with no access to the compromised clients' training procedure or communication with the server.

**Attack mode:** We assume an active attacker with a repeat and adaptive data poisoning on the compromised clients' data. This helps the attack persist over the entire FL training (online attack).

*In summary, the attacker has control of all the data provided to train a local model on compromised clients and can also know the predictions of these clients' local models on any chosen data. However, the attacker can neither interfere with the local model's training process nor poison the model directly. Clients' local training mechanism communicates with the server over an encrypted channel and hence cannot be interfered with.*

### A.3 $(\omega, \rho\chi)$-BYZANTINE RESILIENCE PROOF OF **FLOT**

The below Proposition A.1 signifies that if there are $\rho$ malicious clients, $\chi$ client updates that are trained on highly non-*i.i.d.* data, and the combined validation loss excluding these $\rho + \chi$ model updates is less than $\omega$, then our **FLOT** function is $(\omega, \rho\chi)$ - Byzantine Resilient, where $\omega \geq 0$.

**Proposition A.1** *Let $\mathbb{N} = \{\nabla\mathcal{W}1, \nabla\mathcal{W}2, \ldots, \nabla\mathcal{W}_n\}$ be $n$ total non-i.i.d. set of local clients model updates. Let $\mathbb{R} = \{\nabla\hat{\mathcal{W}}_1, \ldots, \nabla\hat{\mathcal{W}}\rho\}$ be $\rho$ non-i.i.d. set of Byzantine local clients model updates. Let $\mathbb{X} = \{\nabla\hat{\mathcal{W}}'1, \ldots, \nabla\hat{\mathcal{W}}'\chi\}$ be $\chi$ highly non-i.i.d. set of benign local clients model updates. An aggregation rule $\mathcal{A}$ is said to be $(\omega, \rho\chi)$-Byzantine Resilient) if for any $1 \leq \cdots \leq i_1 \cdots \leq i\rho \cdots \leq j_1 \leq \cdots \leq j_\chi \leq \ldots n$, vector*

$$\mathcal{A}(\nabla\mathcal{W}1, \ldots, \nabla\hat{\mathcal{W}}_1, \ldots, \nabla\hat{\mathcal{W}}\rho, \ldots, \nabla\hat{\mathcal{W}}'1, \ldots, \nabla\hat{\mathcal{W}}\chi, \ldots, \nabla\mathcal{W}_n) \tag{16}$$

*satisfies the following*

$$\sum_{\nabla\mathcal{W}_k \in (\mathbb{N}\setminus\mathbb{R})} \mathcal{L}(\mathcal{D}_v, \nabla\mathcal{W}_k) \leq \sum_{\nabla\mathcal{W}_k \in \mathbb{N}} \mathcal{L}(\mathcal{D}_v, \nabla\mathcal{W}_k), \tag{17}$$

$$\sum_{\nabla\mathcal{W}_k \in (\mathbb{N}\setminus\mathbb{X})} \mathcal{L}(\mathcal{D}_v, \nabla\mathcal{W}_k) \leq \sum_{\nabla\mathcal{W}_k \in \mathbb{N}} \mathcal{L}(\mathcal{D}_v, \nabla\mathcal{W}_k), \tag{18}$$

$$\left\| \sum_{\nabla\mathcal{W}_k \in \mathbb{N}} \mathcal{L}(\mathcal{D}_v, \nabla\mathcal{W}_k) - \sum_{\nabla\mathcal{W}_k \in \mathbb{N}\setminus(\mathbb{R}\cup\mathbb{X})} \mathcal{L}(\mathcal{D}_v, \nabla\mathcal{W}_k) \right\| \geq \omega, \tag{19}$$

*for some $\omega \geq 0$. Here, $\mathcal{L}(\mathcal{D}_v, \nabla\mathcal{W}_k)$ denote the validation loss of $\nabla\mathcal{W}_k$ model on validation data $\mathcal{D}_v$. Here, the equality sign in Eq. 17 and Eq. 18 hold true when $\rho = \chi = 0$.*

*Proof.* Without loss of generality, we assume (a) the Byzantine client updates are indexed after benign client vectors, (b) the highly non-*i.i.d.* updates are indexed after the Byzantine updates, i.e.,

$$\textbf{FLOT}(\nabla\mathcal{W}1, \ldots, \nabla\hat{\mathcal{W}}_1, \ldots, \nabla\hat{\mathcal{W}}\rho, \ldots, \nabla\hat{\mathcal{W}}'1, \ldots, \nabla\hat{\mathcal{W}}\chi, \ldots, \nabla\mathcal{W}_n). \tag{20}$$

First, we focus on proving the condition *(i)* (Eq. 17) of Proposition A.1. Consider the first case where $\nabla\mathcal{W}_k \in (\mathbb{N} \setminus \mathbb{R})$, (benign model updates without any malicious updates). Based on the **Theorem 2.** of Jagielski et al. [2018] given by

$$\mathcal{L}_T(\mathcal{D}', \nabla\hat{\mathcal{W}}) \leq \mathcal{L}(\mathcal{D}_{tr}, \nabla\mathcal{W}^*), \tag{21}$$

where $\mathcal{D}'$ represents the malicious training data samples, $\mathcal{D}_{tr}$ is total training data including malicious samples. $\mathcal{L}_T(.,.)$ is the training loss on poisoned $\nabla\hat{\mathcal{W}}$ and main $\nabla\mathcal{W}^*$ models, respectively. However, Jagielski et al. [2018] proved it in terms of data poisoning attacks in centralized machine learning settings with a number of malicious samples under attack. We extend it to federated learning settings in terms of multiple malicious client models that are trained on poisoned and different amounts of non-*i.i.d.* data. Using the set of malicious updates $\mathbb{R}$, set of benign updates $(\mathbb{N} \setminus \mathbb{R}) =$

$\{\nabla\mathcal{W}_1, \nabla\mathcal{W}_2, \ldots, \nabla\mathcal{W}n - \rho\}$, validation data at the server $\mathcal{D}_v$, and Eq. 21, we provide the below formulation using validation loss at the server to prove condition *(i)* of Proposition A.1 as

$$
\begin{aligned}
\mathcal{L}(\mathcal{D}_v, \nabla\mathcal{W}_1) &< \mathcal{L}(\mathcal{D}_v, \nabla\mathcal{W}_1'), \\
\mathcal{L}(\mathcal{D}_v, \nabla\mathcal{W}_2) &< \mathcal{L}(\mathcal{D}_v, \nabla\mathcal{W}_1'), \\
&\cdots \\
\mathcal{L}(\mathcal{D}_v, \nabla\mathcal{W}\rho) &< \mathcal{L}(\mathcal{D}_v, \nabla\mathcal{W}'\rho),
\end{aligned}
\tag{22}
$$

summing up elements on both hand sides and further adding remaining $n - \rho$ elements on both sides and rearranging terms, we get

$$
\sum_{k=1}^{\rho} \mathcal{L}(\mathcal{D}_v, \nabla\mathcal{W}_k) < \sum_{k=1}^{\rho} \mathcal{L}(\mathcal{D}_v, \nabla\mathcal{W}_k'),
\tag{23}
$$

$$
\sum_{k=1}^{\rho} \mathcal{L}(\mathcal{D}_v, \nabla\mathcal{W}_k) + \sum_{k=\rho+1}^{n-\rho} \mathcal{L}(\mathcal{D}_v, \nabla\mathcal{W}_k) < \sum_{k=1}^{\rho} \mathcal{L}(\mathcal{D}_v, \nabla\mathcal{W}_k') + \sum_{k=\rho+1}^{n-\rho} \mathcal{L}(\mathcal{D}_v, \nabla\mathcal{W}_k),
\tag{24}
$$

$$
\sum_{k=1}^{n-\rho} \mathcal{L}(\mathcal{D}_v, \nabla\mathcal{W}_k) < \sum_{k=1}^{\rho} \mathcal{L}(\mathcal{D}_v, \nabla\mathcal{W}_k') + \sum_{k=\rho+1}^{n-\rho} \mathcal{L}(\mathcal{D}_v, \nabla\mathcal{W}_k).
\tag{25}
$$

Adding an additional $\sum_{k=1}^{\rho} \mathcal{L}(\mathcal{D}_v, \nabla\mathcal{W}_k)$ term to the right hand side of Eq. 25 still holds the equation.

$$
\sum_{k=1}^{n-\rho} \mathcal{L}(\mathcal{D}_v, \nabla\mathcal{W}_k) < \sum_{k=1}^{\rho} \mathcal{L}(\mathcal{D}_v, \nabla\mathcal{W}_k') + \sum_{k=\rho+1}^{n-\rho} \mathcal{L}(\mathcal{D}_v, \nabla\mathcal{W}_k) + \sum_{k=1}^{\rho} \mathcal{L}(\mathcal{D}_v, \nabla\mathcal{W}_k),
\tag{26}
$$

$$
\sum_{k=1}^{n-\rho} \mathcal{L}(\mathcal{D}_v, \nabla\mathcal{W}_k) < \sum_{k=1}^{\rho} \mathcal{L}(\mathcal{D}_v, \nabla\mathcal{W}_k') + \sum_{k=1}^{\rho} \mathcal{L}(\mathcal{D}_v, \nabla\mathcal{W}_k) + \sum_{k=\rho+1}^{n-\rho} \mathcal{L}(\mathcal{D}_v, \nabla\mathcal{W}_k),
$$

$$
\sum_{k=1}^{n-\rho} \mathcal{L}(\mathcal{D}_v, \nabla\mathcal{W}_k) < \sum_{k=1}^{n} \mathcal{L}(\mathcal{D}_v, \nabla\mathcal{W}_k),
\tag{27}
$$

$$
\sum_{\nabla\mathcal{W}_k \in (\mathbb{N} \setminus \mathbb{R})} \mathcal{L}(\mathcal{D}_v, \nabla\mathcal{W}_k) \leq \sum_{\nabla\mathcal{W}_k \in \mathbb{N}} \mathcal{L}(\mathcal{D}_v, \nabla\mathcal{W}_k).
\tag{28}
$$

Here $=$ holds true when $\rho = 0$. Finally, Eq. 28 proves the condition *(i)*, i.e., Eq. 17 of Proposition A.1.

Next, we prove the condition *(ii)* (Eq. 18) of Proposition A.1 based on Balakrishnan et al. [2021]. In this work, the authors propose an optimization method to select a subset of client updates that carry representative gradient information of the entire client set. Further, they transmit only the selected subset of client updates to the server for aggregation. The aim is to find an approximation of full clients $(n)$ aggregation gradient via a subset $\mathcal{S}$ of client updates. The authors formulate the problem to provide the upper bound for the aggregated gradient approximation derived from the subset $\mathcal{S}$ of clients as

$$
\left\| \sum_{k \in n} \nabla F_k(v^k) - \sum_{k \in \mathcal{S}} \gamma_k \nabla F_i(v^i) \right\| \leq \sum_{k \in n} \min_{i \in \mathcal{S}} \left\| \nabla F_k(v^k) - \nabla_i F_i(v^i) \right\|,
\tag{29}
$$

where given a subset $\mathcal{S}$, they define a mapping $\sigma : \mathcal{V} \to \mathcal{S}$, such that the gradient information $\nabla F_k(v^k)$ from a client $k$ is approximated by the gradient information from a selected client $\sigma(k) \in \mathcal{S}$. Further, they provide the gradient approximation error as

$$
\left\| \frac{1}{n} \sum_{k \in \mathcal{S}^t} \gamma_k \nabla F_k(v_t^k) - \frac{1}{n} \sum_{k \in n} \nabla F_k(v_t^k) \right\| \leq \delta,
$$
$$
\left\| \sum_{k \in \mathcal{S}^t} \gamma_k \nabla F_k(v_t^k) - \sum_{k \in n} \nabla F_k(v_t^k) \right\| \leq n\delta,
$$
(30)

where $t$ is the communication round, $\{\gamma\}_{k \in \mathcal{S}_t}$ are the weights assigned to gradients, and $\delta$ is the error rate that is used as a measure to characterize the goodness of gradient approximation. The above equation states that the gradient approximation from subset $\mathcal{S}$ of clients at communication round $t$ is less than $n\delta$ times full gradient aggregation from all clients. Further, we use this observation and extend it to validation loss that there exists a subset of client updates $(\mathbb{N} \setminus \mathbb{X})$ whose sum of validation losses is less than that of the sum of total clients. It is given as

$$
\left\| \sum_{k \in \mathcal{S}^t} \mathcal{L}(\mathcal{D}_v, v_t^k) - \sum_{k \in n} \mathcal{L}(\mathcal{D}_v, v_t^k) \right\| \leq n\delta,
$$
$$
\left\| \sum_{\nabla \mathcal{W}_k \in (\mathbb{N} \setminus \mathbb{X})} \mathcal{L}(\mathcal{D}_v, \nabla \mathcal{W}_k) - \sum_{\nabla \mathcal{W}_k \in \mathbb{N}} \mathcal{L}(\mathcal{D}_v, \nabla \mathcal{W}_k) \right\| \leq n\delta.
$$
(31)

Here, $\mathbb{N} \setminus \mathbb{X}$ denote the subset of clients obtained after discarding $\chi$ non-*i.i.d.* clients whose validation loss is higher than that of remaining clients. Finally, the below equation proves the condition *(ii)*, i.e., Eq. 18 of Proposition A.1.

$$
\sum_{\nabla \mathcal{W}_k \in (\mathbb{N} \setminus \mathbb{X})} \mathcal{L}(\mathcal{D}_v, \nabla \mathcal{W}_k) \leq n\delta \sum_{\nabla \mathcal{W}_k \in \mathbb{N}} \mathcal{L}(\mathcal{D}_v, \nabla \mathcal{W}_k),
$$
$$
\sum_{\nabla \mathcal{W}_k \in (\mathbb{N} \setminus \mathbb{X})} \mathcal{L}(\mathcal{D}_v, \nabla \mathcal{W}_k) \leq \sum_{\nabla \mathcal{W}_k \in \mathbb{N}} \mathcal{L}(\mathcal{D}_v, \nabla \mathcal{W}_k).
$$
(32)

Combining Eq. 28 and Eq. 32 we get

$$
\sum_{\nabla \mathcal{W}_k \in (\mathbb{N} \setminus \mathbb{R})} \mathcal{L}(\mathcal{D}_v, \nabla \mathcal{W}_k) \leq \sum_{\nabla \mathcal{W}_k \in \mathbb{N}} \mathcal{L}(\mathcal{D}_v, \nabla \mathcal{W}_k),
$$
(33)

$$
\sum_{\nabla \mathcal{W}_k \in (\mathbb{N} \setminus \mathbb{X})} \mathcal{L}(\mathcal{D}_v, \nabla \mathcal{W}_k) \leq \sum_{\nabla \mathcal{W}_k \in \mathbb{N}} \mathcal{L}(\mathcal{D}_v, \nabla \mathcal{W}_k),
$$
(34)

$$
\sum_{\nabla \mathcal{W}_k \in (\mathbb{N} \setminus \mathbb{R})} \mathcal{L}(\mathcal{D}_v, \nabla \mathcal{W}_k) + \sum_{\nabla \mathcal{W}_k \in \mathbb{N} \setminus \mathbb{X}} \mathcal{L}(\mathcal{D}_v, \nabla \mathcal{W}_k) \leq \sum_{\nabla \mathcal{W}_k \in \mathbb{N}} \mathcal{L}(\mathcal{D}_v, \nabla \mathcal{W}_k) + \sum_{\nabla \mathcal{W}_k \in \mathbb{N}} \mathcal{L}(\mathcal{D}_v, \nabla \mathcal{W}_k),
$$
(35)

$$
\sum_{\nabla \mathcal{W}_k \in \mathbb{R}} \mathcal{L}(\mathcal{D}_v, \nabla \mathcal{W}_k) + \sum_{\nabla \mathcal{W}_k \in \mathbb{X}} \mathcal{L}(\mathcal{D}_v, \nabla \mathcal{W}_k) + 2 \sum_{\nabla \mathcal{W}_k \in \mathbb{N} \setminus (\mathbb{R} \cup \mathbb{X})} \mathcal{L}(\mathcal{D}_v, \nabla \mathcal{W}_k) \leq 2 \sum_{\nabla \mathcal{W}_k \in \mathbb{N}} \mathcal{L}(\mathcal{D}_v, \nabla \mathcal{W}_k),
$$
(36)

$$
2 \sum_{\nabla \mathcal{W}_k \in \mathbb{N} \setminus (\mathbb{R} \cup \mathbb{X})} \mathcal{L}(\mathcal{D}_v, \nabla \mathcal{W}_k) \leq 2 \sum_{\nabla \mathcal{W}_k \in \mathbb{N}} \mathcal{L}(\mathcal{D}_v, \nabla \mathcal{W}_k) - \sum_{\nabla \mathcal{W}_k \in \mathbb{R}} \mathcal{L}(\mathcal{D}_v, \nabla \mathcal{W}_k) - \sum_{\nabla \mathcal{W}_k \in \mathbb{X}} \mathcal{L}(\mathcal{D}_v, \nabla \mathcal{W}_k),
$$
(37)

$$2 \sum_{\nabla \mathcal{W}_k \in \mathbb{N} \setminus (\mathbb{R} \cup \mathbb{X})} \mathcal{L}(\mathcal{D}_v, \nabla \mathcal{W}_k) \leq \sum_{\nabla \mathcal{W}_k \in \mathbb{N}} \mathcal{L}(\mathcal{D}_v, \nabla \mathcal{W}_k) + \sum_{\nabla \mathcal{W}_k \in \mathbb{R}} \mathcal{L}(\mathcal{D}_v, \nabla \mathcal{W}_k) + \sum_{\nabla \mathcal{W}_k \in \mathbb{X}} \mathcal{L}(\mathcal{D}_v, \nabla \mathcal{W}_k) +$$
$$\sum_{\nabla \mathcal{W}_k \in \mathbb{N} \setminus (\mathbb{R} \cup \mathbb{X})} \mathcal{L}(\mathcal{D}_v, \nabla \mathcal{W}_k) - \sum_{\nabla \mathcal{W}_k \in \mathbb{R}} \mathcal{L}(\mathcal{D}_v, \nabla \mathcal{W}_k) - \sum_{\nabla \mathcal{W}_k \in \mathbb{X}} \mathcal{L}(\mathcal{D}_v, \nabla \mathcal{W}_k), \tag{38}$$

$$2 \sum_{\nabla \mathcal{W}_k \in \mathbb{N} \setminus (\mathbb{R} \cup \mathbb{X})} \mathcal{L}(\mathcal{D}_v, \nabla \mathcal{W}_k) \leq \sum_{\nabla \mathcal{W}_k \in \mathbb{N}} \mathcal{L}(\mathcal{D}_v, \nabla \mathcal{W}_k) + \sum_{\nabla \mathcal{W}_k \in \mathbb{N} \setminus (\mathbb{R} \cup \mathbb{X})} \mathcal{L}(\mathcal{D}_v, \nabla \mathcal{W}_k), \tag{39}$$

$$\sum_{\nabla \mathcal{W}_k \in \mathbb{N} \setminus (\mathbb{R} \cup \mathbb{X})} \mathcal{L}(\mathcal{D}_v, \nabla \mathcal{W}_k) \leq \sum_{\nabla \mathcal{W}_k \in \mathbb{N}} \mathcal{L}(\mathcal{D}_v, \nabla \mathcal{W}_k), \tag{40}$$

$$\left\| \sum_{\nabla \mathcal{W}_k \in \mathbb{N}} \mathcal{L}(\mathcal{D}_v, \nabla \mathcal{W}_k) - \sum_{\nabla \mathcal{W}_k \in \mathbb{N} \setminus (\mathbb{R} \cup \mathbb{X})} \mathcal{L}(\mathcal{D}_v, \nabla \mathcal{W}_k) \right\| \geq 0, \tag{41}$$

generalizing,

$$\left\| \sum_{\nabla \mathcal{W}_k \in \mathbb{N}} \mathcal{L}(\mathcal{D}_v, \nabla \mathcal{W}_k) - \sum_{\nabla \mathcal{W}_k \in \mathbb{N} \setminus (\mathbb{R} \cup \mathbb{X})} \mathcal{L}(\mathcal{D}_v, \nabla \mathcal{W}_k) \right\| \geq \omega, \tag{42}$$

where $\omega \geq 0$. Finally, Eq. 42 proves the condition *(iii)*, i.e., Eq 19 of Proposition A.1.

### A.4 More Results on Hypothesis Testing

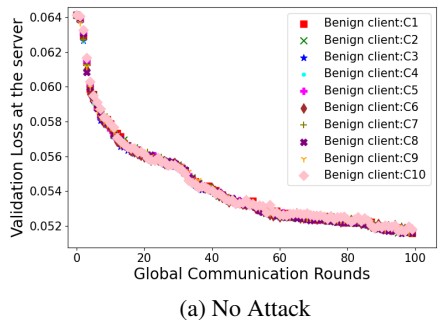 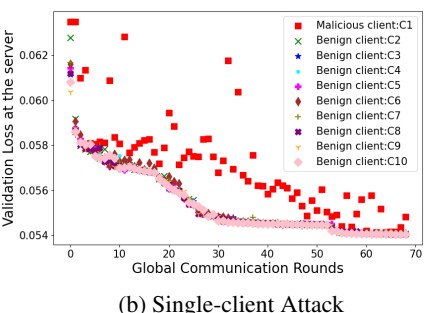

(a) No Attack        (b) Single-client Attack

Figure 4: Validation losses of individual client model at the server *w.r.t.* global communication rounds under no attack and single-client attack settings for KBTS dataset. Here, the global model is updated with the remaining good-performing client updates. For the next iteration, the clients train their local models using this new global model.

Our defense is based on the hypothesis that the updates from a malicious client doing data poisoning will differ from benign client updates in terms of loss of validation data at the server. Figure. 4 shows the validation loss of 10 clients under no attack and single-client attack settings. We observe under no attack settings, the validation loss of all the updates is clustered together (shows similar behavior) and reduces with the increase in global communication rounds. On the contrary, there is a clear dispersion in the malicious client (C1-squared entries) loss values compared to other benign clients' losses. Hence, our **FLOT** used loss function-based model rejection to suppress updates from malicious clients.

*Test loss analysis:* Figure 5 shows the performance of **FLOT** compared to other Byzantine server rules. For brevity, we showed the hard case of a multi-client attack (33% Byzantine) for the KBTS dataset with ten clients. We observed that our **FLOT** showed a lower loss, followed by Krum. Further,

we observe that **FLOT** configuration is better in this case as **FLOT+RS** randomly selects some clients and applies **FLOT** on top of it. As there are less number of clients for the KBTS dataset, sampling clients randomly and using **FLOT** leads to losing the benign client updates and lower performance. Trimmed mean, with its ability to trim client updates from beginning to end, leads to discarding benign updates and including malicious updates. Hence, it performs worst compared to other methods.

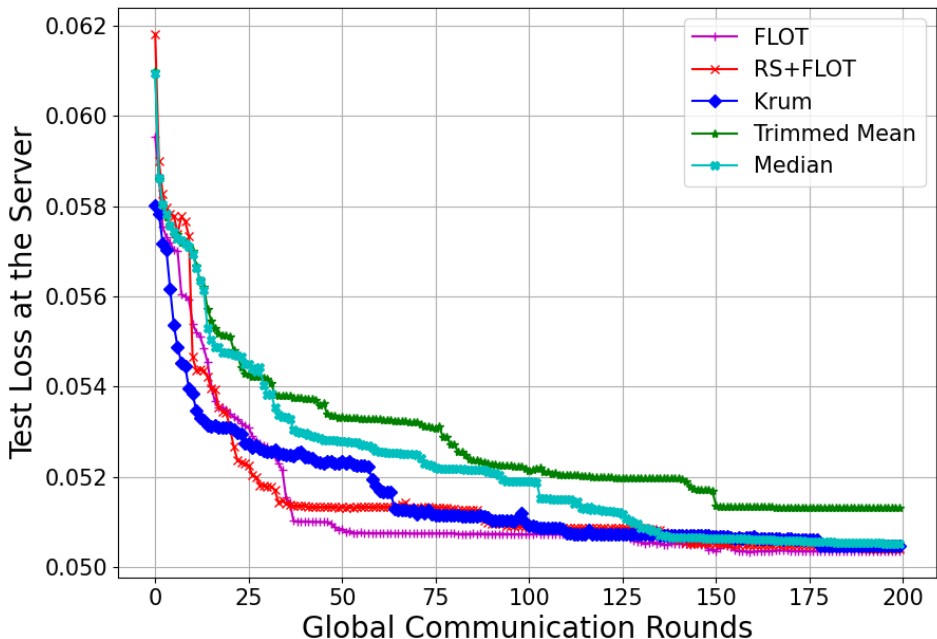

Figure 5: Comparison of test losses of **FLOT** with different Byzantine aggregation techniques at the server for 200 global communication rounds under 33% multi-attack settings for the KBTS dataset.

### A.5 ADDITION EXPERIMENTAL DETAILS

**Datasets and implementation details. GTSRB** [Stallkamp et al., 2011] is a well-known benchmark dataset for traffic sign classification. It consists of 43 traffic sign classes. Most (70%) of the training data (27,446 samples) is divided using the Dirichlet distribution with $\alpha = 1$. Further, 10% (3920 samples) is used as validation data at the server, and the remaining 20% of the data (7842 samples) is used for testing. **KUL Belgium traffic sign (KBTS)** dataset [Mathias et al., 2013] is another benchmark dataset for traffic sign classification. It consists of 62 traffic sign classes. A majority (70%) of the training data (4884 samples) is divided using the Dirichlet distribution with $\alpha = 1$. Further, 10% (697 samples) is used as validation data, and the remaining 20% of the data (1397 samples) is used for testing. **CIFAR10** [Krizhevsky et al., 2009] is a well-known benchmark dataset for classification that contains 60,000 samples with ten different classes. Most (70%) of the training data (42,000 samples) is divided using the Dirichlet distribution with $\alpha = 1$. Further, 10% (6000 samples) is used as validation data, and the remaining 20% of the data (12000 samples) is used for testing. Finally, EMNIST [Cohen et al., 2017] is another benchmark dataset of 671,585 samples of handwritten characters & digits with 62 classes, including upper and lowercase handwritten characters. Further, we consider 10,000 total clients, out of which 100 client updates are randomly selected at every communication round. 4,70,000 samples

Table 4: CNN configuration

| Black-box CNN (4 Conv layers) |
| --- |
| input ($150 \times 150$ RGB images) |
| conv2d_64; kernel 5; stride 1 |
| conv2d_128; kernel 3; stride 1 |
| conv2d_256; kernel 1; stride 1 |
| conv2d_256; kernel 1; stride 1 |
| Fully connected layer 1 |
| Fully connected layer 2 |
| Softmax classifier |

are divided using the Dirichlet distribution with $\alpha = 1$. Further, 5000 samples are used as validation data, and 10000 samples are used for testing.

**Classifier architectures.** We built a custom 4-layer CNN architecture followed by two fully connected layers and treated this as a global model, as shown in Table 4. We experiment with GTSRB, KBTS, and CIFAR10 datasets using this architecture. The model is trained with images of size $150 \times 150$ using categorical cross-entropy as loss function optimized using Adam optimizer. Additionally, we use ResNet18 [He et al., 2015] and LeNet [LeCun et al., 1998] architecture that takes an input of size $224 \times 224$ and $32 \times 32$, respectively, for CIFAR10 and EMNIST datasets. During the training of the global classifier for 200 epochs through FL protocol, each client trains for $E = 5$ local epochs on the local data with a batch size $b_s = 64$ and with a learning rate of $l_r = 0.01$.

All the clients are trained individually and sequentially at each global epoch. We used Python 3.6+, Pytorch, and Python OT (especially `ot.lp.barycenter` function with solver='interior-point') and implemented the entire setup on Nvidia Tesla M60 GPU & 8GB RAM.

**Baselines.** We have chosen to compare **FLOT** with relevant baselines commonly used in the literature. We believe these baselines provide a fair evaluation of **FLOT**'s performance in defending against untargeted data poisoning attack scenarios.

1. *FedAvg [McMahan et al., 2017]:* Normal federated learning without any defense. Ideally, **FLOT** should perform similarly to this baseline under **no attack** scenarios.

2. *Random Sampling (RS) of the Clients:* This represents the FL system with random sampling, where the server randomly selects some updates for aggregation. As our **FLOT** involves generating loss function-based weighted coefficients that drop the malicious clients, followed by OT optimization, it should perform better than RS.

3. *Power-of-choice [Cho et al., 2020]:* In this work, the server selects the clients with the largest training losses.

4. *DivFL [Balakrishnan et al., 2021]:* This is a recent work that proposes a technique to perform FL by selecting a group of clients based on submodular optimization.

5. *FLOT Configurations:* We use two configurations of **FLOT**, namely, **FLOT** (our method) and **FLOT+RS** (our method includes random sampling for better results).

We use the following Byzantine Robust Aggregation approaches to perform a comparative evaluation:

1. *Krum [Blanchard et al., 2017]:* Krum selects one local model updates that are representative of a majority of client models. We set $c = 10$ for the GTSRB and CIFAR10 datasets and $c = 3$ for the KBTS dataset to handle the 33% malicious clients in our experimentation.

2. *TM [Yin et al., 2018]:* Trimmed mean (TM) aggregates each dimension of input updates separately and sorts the values along the $i^{th}$-dimension. Then, it removes $x$ largest and smallest values of that dimension and computes the average of the rest. We consider the suggested configuration of $x = 5$ for GTSRB, CIFAR10, and $x = 1$ for KBTS datasets to handle the 33% malicious clients in our experimentation.

3. *Median [Yin et al., 2018]:* The median aggregates each dimension of input updates separately and sorts the values of the $i^{th}$-dimension. Then, it takes the median as the global model's $i^{th}$ parameter.

Finally, we use the below recent FL defense methods for our evaluation.

1. *FLTrust [Cao et al., 2021]:* In this method, the server trains an auxiliary model using a root dataset and computes trust scores for clients based on the similarity of their weight updates to the server model. The server then updates the global model by taking a weighted average of the client models, with the weights proportional to their trust scores.

2. *LoMar [Li et al., 2023]:* This is a recent defense method which uses a two phase method. It scores model updates using kernel density estimation in the first phase and determines an optimal threshold to distinguish between malicious and clean updates in the second phase.

3. *FL-Defender [Jebreel & Domingo-Ferrer, 2023]:* This is another recent defense method. It analyzes the behaviour of neurons related to the attacks and proposes robust discriminative

features using worker-wise angle similarity. Then, it compresses similarity vectors and re-weights worker updates before aggregation.

**Non-*i.i.d.* data distribution in FL.** The influence of varying non-*i.i.d.* data distribution is a critical aspect that warrants further exploration. This examination allows us to better understand the interplay between the Dirichlet distribution parameter $\beta$ and the resulting data distribution characteristics. The relationship between $\beta$ and the sample data partition is pivotal in comprehending the behavior of our experimental setup.

The Dirichlet distribution [Minka, 2000] is a fundamental probabilistic model used in FL to characterize the distribution of data across different clients. This distribution is controlled by a parameter $\beta$, which plays a pivotal role in influencing the degree of non-*i.i.d.*ness in the dataset distribution. The working principle of the Dirichlet distribution involves generating data partitions across clients based on their unique characteristics. The mathematical formulation of the Dirichlet distribution is expressed as follows:

$$p(x_1, x_2, \ldots, x_K | \beta) = \frac{1}{B(\beta)} \prod_{i=1}^{K} x_i^{\beta_i - 1},$$

where $x_1, x_2, \ldots, x_K$ represent the proportions of data allocated to each client. $K$ is the total number of classes. $\beta = (\beta_1, \beta_2, \ldots, \beta_K)$ is a vector of parameters that influence the distribution (in our approach, we consider a case where all the $\beta_i$ values to be the same, resulting in a symmetric Dirichlet distribution). $B(\beta)$ represents the multivariate Beta function, which serves as a normalizing constant in the probability density function of the Dirichlet distribution. This function ensures that the calculated probabilities from the distribution sum up to 1 over the simplex defined by the data proportions.

The formula for the multivariate Beta function $B(\beta)$ is given by:

$$B(\beta) = \frac{\prod_{i=1}^{K} \Gamma(\beta_i)}{\Gamma(\sum_{i=1}^{K} \beta_i)}.$$

Through manipulation of the parameter $\beta$, the density of independently and identically distributed (*i.i.d.*) data splits among clients can be shaped, thereby determining the non-*i.i.d.* nature of the data distribution. Larger values of $\beta$ lead to a more uniformly distributed data landscape among clients, effectively reducing variability in their data distributions. Conversely, smaller values of $\beta$ result in a more concentrated or skewed data distribution, consequently introducing varying degrees of heterogeneity and non-*i.i.d.*ness among clients. Proper calibration of $\beta$ becomes essential for FL systems, allowing them to account for the inherent heterogeneity in real-world client data, a crucial factor for model robustness and generalization.

Our experimentation delves into the symbiotic relationship between the Dirichlet distribution parameter $\beta$ and FL attack dynamics. This interaction is pivotal for our study, as non-*i.i.d.* client datasets can significantly impact the global model's accuracy, even prior to the introduction of an attack. This pre-existing effect arises due to biased and overfitted client models that emerge from non-*i.i.d.* local datasets. This phenomenon amplifies the overall attack impact and elevates the robustness of a defense method.

However, it's important to recognize that the impact of non-*i.i.d.*ness is not solely governed by $\beta$. A confluence of factors, such as the total number of clients, the clients selected per round, and local and global training epochs, collectively influence the magnitude of the GTA under no attack scenarios. In conclusion, our in-depth analysis of the non-*i.i.d.* data distribution's impact on FL attacks provides vital insights into the complex dynamics governing FL system performance. The careful calibration of $\beta$ and its repercussions on data distribution elucidate the underlying factors that can lead to substantial variations in model accuracy and **FLOT** effectiveness. This exploration enriches our understanding of FL's behavior under varying conditions and underscores the importance of accounting for non-*i.i.d.*ness in practical scenarios.

### A.6 MORE EMPIRICAL ANALYSIS AND ABLATION STUDIES

**Robustness against different attacks.** We have evaluated **FLOT** using the below attacks.

1. *M-SimBA Kumar et al. [2020]:* This is another centralized ML black-box data poisoning attack. It uses randomized gradients similar to SimBA but tries to reduce the loss of the most confused class that the model misclassifies a sample with the highest probability.

2. *DPA-SLF Shejwalkar et al. [2022]:* This is a data poisoning -static label flipping attack, where each compromised client flips the labels of their data from true label $y \in [0, C-1]$ to false label $(C - 1 - y)$ if $C$ is even and to false label $(C - y)$ if $C$ is odd, where $C$ is the number of classes.

3. *DPA-DLF Shejwalkar et al. [2022]:* This is another data poisoning -dynamic label flipping attack that uses a surrogate model benign data (standard FL model) and flip label $y$ to the least probable label it generates for a given sample. We use the same model architectures as a surrogate for the respective datasets.

Table 5 presents the attack success rates of three distinct attacks under our FL setup with no defense and the aforementioned maliciousness levels, namely the single-client attack (1A), as well as the multi-client attack with 10%, 20%, 30%, 40%, and 50% maliciousness for EMNIST dataset. The attack success rate is defined as the ratio of misclassified test samples to the total number of samples at the server under that specific attack setting. Our analysis reveals that the black-box gradient noise data poisoning attack MSimBA outperforms the dynamic and static label flip attacks in the FL setup in terms of attack success rate under no defense.

Table 5: Attack success rate ($\uparrow$) of MSimBA, DPA-SLF, and DPA-DLF attacks for different attack percentages on EMNIST dataset without defense.

| Attack percentage (%) | MSimBA | DPA-SLF | DPA-DLF |
|---|---|---|---|
| 1A | **0.16** | 0.15 | 0.15 |
| 20 | **0.27** | 0.21 | 0.23 |
| 30 | **0.26** | 0.20 | 0.24 |
| 40 | **0.53** | 0.49 | 0.51 |
| 50 | **0.77** | 0.53 | 0.69 |

Additionally, we conducted an ablation study to evaluate the performance of our **FLOT** framework in comparison to other defense mechanisms, including FedAvg, DivFL, Krum, LoMar, and FLDefender, under DPA-SLF and DPA-DLF attacks. The results, as presented in Table 6 and Table 7, showcase the superior performance of our **FLOT** method, with an approximate 1-4% higher accuracy compared to the other methods. It's worth noting that our approach exhibits higher robustness against DPA-SLF, a static label flip attack, in comparison to DPA-DLF, a dynamic label flip attack. In summary, our OT-based dynamic update discarding mechanism consistently preserves the GTA more effectively than other methods under DPA-SLF and DPA-DLF attacks, demonstrating its robustness and adaptability across a wide range of attack strategies.

Table 6: GTA% ($\uparrow$) performance comparison of **FLOT** method under DPA-SLF (Shejwalkar et al. [2022]) attack for CIFAR10 and EMNIST datasets.

| Defense Method | CIFAR10 | | EMNIST | |
|---|---|---|---|---|
| | 1A | 50% | 1A | 50% |
| FedAvg | 87.18 | 49.36 | 84.14 | 49.38 |
| DivFL | 84.12 | 61.26 | 85.92 | 61.48 |
| Krum | 86.95 | 68.64 | 85.60 | 65.76 |
| LoMar | 87.34 | 73.39 | 86.12 | 67.73 |
| FLDefender | 88.75 | 74.83 | 86.31 | 69.64 |
| **FLOT (ours)** | **89.36** | **78.12** | **87.71** | **72.33** |

Under the no-attack setting, our approach closely performed to that of the FL baseline with $< 1\%$ difference for the GTSRB and KBTS dataset and outperformed the CIFAR10 dataset, as shown in Table 8. This is due to a large number of classes with inter and intra-class variability in the GTSRB and KBTS dataset that led to the discarding of benign client models with a slight difference in the loss values. Also, the FedAvg tries to achieve the local optimum error rate when the objective function is strongly convex under no attack. On the contrary, given a good amount of data, our **FLOT** configuration was able to sample updates that improved performance under no attack on the CIFAR10 dataset.

Table 7: GTA% ($\uparrow$) performance comparison of **FLOT** method under DPA-DLF (Shejwalkar et al. [2022]) attack for CIFAR10 and EMNIST datasets.

| Defense Method | CIFAR10 | | EMNIST | |
|---|---|---|---|---|
| | 1A | 50% | 1A | 50% |
| FedAvg | 85.32 | 51.91 | 85.48 | 47.62 |
| DivFL | 83.68 | 63.11 | 85.05 | 62.16 |
| Krum | 84.65 | 65.62 | 84.30 | 63.45 |
| LoMar | 86.17 | 71.36 | 86.15 | 69.54 |
| FLDefender | 86.05 | 75.42 | 85.81 | 67.28 |
| **FLOT (ours)** | **87.65** | **79.37** | **86.53** | **71.36** |

**Multi-client attack + defense analysis.** We extended our evaluation of **FLOT** configurations to include a 33% MSimBA multi-client attack scenario as an extension to the main paper results. Our findings, as presented in Table 9, consistently demonstrate the superior performance of **FLOT** configurations, with an accuracy improvement of approximately over 1% compared to other methods.

Furthermore, to showcase the versatility and adaptability of **FLOT** across different model architectures, we evaluated its performance on the CIFAR10 dataset using the ResNet18 architecture under a

Table 8: GTA% (↑) for no attack and defense case.

| Defense Method | GTSRB | KBTS | CIFAR10 |
|---|---|---|---|
| FedAvg | **87.8** | 90.02 | 91.23 |
| RS | 86.68 | 87.92 | 90.54 |
| PC | 87.56 | 88.05 | 92.64 |
| DivFL | 87.12 | **89.96** | **92.86** |
| Krum | 86.72 | 89.97 | 91.46 |
| TM | 84.32 | 88.52 | 90.64 |
| Median | 85.23 | 88.27 | 89.91 |
| LoMar | 85.12 | 88.12 | 89.62 |
| FLDefender | 86.28 | 89.12 | 91.51 |
| **FLOT (ours)** | 86.24 | 89.12 | 91.51 |
| **FLOT+RS (ours)** | 87.01 | 89.36 | 92.37 |

Table 9: GTA% (↑) for multi-client MSimBA attack (33%) and defense case.

| Defense Method | GTSRB | KBTS | CIFAR10 |
|---|---|---|---|
| FedAvg | 70.63 | 83.26 | 85.03 |
| RS | 65.45 | 84.24 | 82.98 |
| PC | 63.72 | 80.27 | 73.86 |
| DivFL | 72.08 | 81.63 | 74.12 |
| Krum | 79.98 | 84.29 | 85.12 |
| TM | 77.45 | 84.09 | 84.43 |
| Median | 78.64 | 84.97 | 83.36 |
| LoMar | 79.28 | 83.36 | 84.15 |
| FLDefender | 80.15 | 84.92 | 84.96 |
| **FLOT (ours)** | 81.12 | **85.94** | 85.21 |
| **FLOT+RS (ours)** | **82.26** | 85.02 | **86.24** |

33% multi-client attack generated by MSimBA. Our results, as displayed in Table 10, highlight the significant advantage of our **FLOT** approach, outperforming other methods by approximately 3

Lastly, to emphasize the scalability of **FLOT** in handling multi-client attacks, we conducted evaluations across various attack percentages (ranging from 10% to 40%) using the EMNIST dataset. Remarkably, **FLOT** consistently outperformed other methods across all attack scenarios, as demonstrated in Table 11. These results underline the effectiveness and robustness of our **FLOT** method in diverse and challenging multi-client attack settings.

**FLOT Runtime analysis.** In our final evaluation, we focused on assessing the runtime performance of our **FLOT** method. We considered two scenarios: the best-case scenario involving ten clients for the KBTS dataset and the worst-case scenario with 100 clients for the EMNIST dataset. Our observations indicate that there is no significant increase in runtime when utilizing **FLOT**, with execution times remaining close to those of standard FL procedures. Interestingly, we even observed a reduction in runtime when implementing **FLOT** in conjunction with random sampling (**FLOT+RS**), as illustrated in Table 12. These results underscore the practical efficiency of our **FLOT** method, as it demonstrates comparable runtime to traditional FL processes, making it easily integrated into current FL systems.

Table 10: GTA% (↑) for multi-client MSimBA attack (33%) and using ResNet18 on CIFAR10 dataset.

| Defense Method | 33% |
|---|---|
| FedAvg | 71.34 |
| DivFL | 65.24 |
| Krum | 77.14 |
| LoMar | 78.31 |
| FLDefender | 77.92 |
| **FLOT (ours)** | **81.62** |

Table 11: GTA% (↑) for multi-client MSimBA attack (10, 20, 30, 40)% on EMNIST dataset.

| Defense Method | 10% | 20% | 30% | 40% |
|---|---|---|---|---|
| FedAvg | 76.19 | 56.24 | 49.37 | 35.33 |
| DivFL | 80.32 | 69.26 | 54.82 | 42.31 |
| Krum | 81.36 | 75.10 | 68.08 | 46.68 |
| LoMar | 82.71 | 78.61 | 73.40 | 65.70 |
| FLDefender | 83.55 | 80.37 | 76.23 | 63.67 |
| **FLOT (ours)** | **84.42** | **81.38** | **78.27** | **69.78** |

Table 12: Execution runtime (seconds ↓) of different defense methods for best-case ten clients (10) for KBTS dataset and worst-case hundred clients (100C) for EMNIST dataset.

| Defense Method | Best-case (10C) | Worst-case (100C) |
|---|---|---|
| FedAvg | 350 | 730 |
| RS | 350 | 730 |
| PC | 410 | 830 |
| DivFL | 430 | 850 |
| Krum | 430 | 850 |
| TM | 350 | 730 |
| Median | 340 | 710 |
| FLTrust | 400 | 810 |
| LoMar | 390 | 780 |
| FLDefender | 400 | 810 |
| **FLOT (ours)** | 390 | 780 |
| **FLOT+RS (ours)** | 360 | 740 |

