# OpenReview forum: "Byzantine-Robust Dynamic Weighted Aggregation Framework for Optimal Attack Mitigation in Federated Learning"
_ICLR.cc/2024/Conference — ICLR 2024 Conference Withdrawn Submission_

### Official Review · Reviewer_eNX5 · 2023-10-27

**Soundness:** 2 fair
**Presentation:** 1 poor
**Contribution:** 2 fair
**Rating:** 3
**Confidence:** 4

**Summary:**

In this paper, the authors propose a novel framework called Federated Learning Optimal Transport (FLOT), the time complexity of which is $O(n \log(n)d)$. The authors provide the convergence analysis of FLOT for convex problems under non-i.i.d. settings. Empirical results on four different datasets verify the effectiveness of FLOT.

**Strengths:**

1. The idea of using the optimal transport framework in Byzantine-robust FL is novel and interesting.
2. I appreciate that the authors empirically test different methods on four different datasets.

**Weaknesses:**

Weakness 1. The expression of section 6 is unclear. Specifically,

a) The assumptions are mentioned only when they are used. I strongly suggest the authors explicitly list the assumptions at the beginning of the convergence analysis.

b) The authors say that they analyze the convergence for convex problems (line 265). However, it is unclear where the convexity is used.

c) It is unclear whether the used PL inequality (line 274) is an assumption or is just derived from the convexity.

d) It is confusing what the notations $(\cdot)$# (line 273) and $\lessapprox$ (line 278) mean.

Weakness 2. The convergence analysis uses both the smoothness (line 275) and the convexity (line 265) assumptions, which are quite strong. In addition, the analysis is for gradient descent instead of the more frequently used stochastic gradient descent.

Weakness 3. The authors claim that the time complexity of FLOT is smaller than Krum. However, it will be more meaningful to directly compare the empirical running time since the validation operation in FLOT is time-consuming (which is not required in Krum).

Weakness 4. There are some omitted baselines. For example, Zeno [1] also uses the validation loss to obtain robustness. Besides, there are also some existing Byzantine-robust learning methods for non-i.i.d. cases, such as nearest neighbor mixing [2] and Byz-VR-MARINA [3].

[1] Cong Xie et al. Zeno: Distributed Stochastic Gradient Descent with Suspicion-based Fault-tolerance. ICML 2019.

[2] Youssef Allouah et al. Fixing by Mixing: A Recipe for Optimal Byzantine ML under Heterogeneity. AISTATS 2023.

[3] Eduard Gorbunov et al. Variance Reduction is an Antidote to Byzantine Workers: Better Rates, Weaker Assumptions and Communication Compression as a Cherry on the Top. ICLR 2023.

**Questions:**

Please see the weaknesses.

---

### Official Review · Reviewer_CP8z · 2023-10-29

**Soundness:** 2 fair
**Presentation:** 3 good
**Contribution:** 2 fair
**Rating:** 3
**Confidence:** 4

**Summary:**

The paper presents FLOT, a dynamic weighted federated aggregation method that uses optimal transport techniques to defend against untargeted data poisoning attacks in the Federated Learning (FL) framework. FLOT achieves this by interpolating global model updates using loss-based weighted coefficients and employing Wasserstein barycenters for OT optimization to obtain a smoother global model, effectively filtering out malicious updates.

**Strengths:**

1. This paper provides a convergence analysis of FLOT and provide a theoretical analysis of the attack.

2. The authors conduct extensive experiments on multiple datasets under IID and non-IID settings.

**Weaknesses:**

1. Even though the authors cite several papers to show that previous works also assume there is a validation dataset on the server, it is still not convincing that the server should have a validation dataset in IID with the global dataset. If the server has such an inclusive and high-quality validation dataset, I do not see any challenge in defending the poisoning attack.

2. Even though the authors provide a theoretical analysis of the convergence of FLOT, the theoretical robustness of FLOT against the attacks is missed.

3. The authors do not discuss the robustness of the adaptive attack, where the attackers have prior information about the defense.

4. Some important baselines for improving robustness against poisoning attacks vis attacker detection are missed, such as [1].

[1] Zhang, Zaixi, et al. "FLDetector: Defending federated learning against model poisoning attacks via detecting malicious clients." Proceedings of the 28th ACM SIGKDD Conference on Knowledge Discovery and Data Mining. 2022.

**Questions:**

Please see the weaknesses.

---

### Official Review · Reviewer_eeKh · 2023-10-31

**Soundness:** 2 fair
**Presentation:** 3 good
**Contribution:** 2 fair
**Rating:** 3
**Confidence:** 4

**Summary:**

This paper proposes to an optimal transport-based dynamic weighted federated aggregation method designed to mitigate untargeted data poisoning attacks within the FL framework. The theoretical convergence results are provided. However, the proposed method relies on validation data at the server. Some advanced defense methods are missing in the introduction and experiments.

**Strengths:**

1. This paper proposes an optimal transport-based dynamic weighted federated aggregation method designed to mitigate untargeted data poisoning attacks within the FL framework.
2. The complexity of the proposed defense method has a substantial improvement compared to Krum.
3. The theoretical convergence results are provided.
4. Eextensive experimental results demonstrate that the proposed method consistently outperform other methods, including Byzantine robust aggregation rules, in terms of classification performance under both single-client and 50% Byzantine worker scenarios.

**Weaknesses:**

1. The proposed method relies on validation data at the server. This validation data will affect the performance of the proposed defense method much. If the server cannot obtain the training data, the performance of the proposed method should be shown.
2. Some advanced defense methods are missing in the introduction and experiments, e.g., [1] and [2].
3. The datasets used in this paper are too simple and limited to image classification tasks. Some real-world datasets, e.g., imagenet and cifar100, and other tasks should be considered, e.g., sentiment classification and multi-modal tasks.
4. The federated self-supervised learning framework should be considered in the experiments.
5. The convergence analysis is not meaningful, due to some impractical assumptions, e.g., Polyak-Łojasiewicz (PL) inequality.
6. The attacker setting in this paper is also simple. There are some advanced model poisoning attack methods, e.g., [3] and etc.

The references mentioned are as follows:
[1] Xiaoyu Cao, Jinyuan Jia, Zaixi Zhang, and Neil Zhenqiang Gong. "FedRecover: Recovering from Poisoning Attacks in Federated Learning using Historical Information," In IEEE Symposium on Security and Privacy, 2023.
[2] Sungwon Park, Sungwon Han, Fangzhao Wu, Sundong Kim, Bin Zhu, Xing Xie, and Meeyoung Cha. "FedDefender: Client-Side Attack-Tolerant Federated Learning," In Proceedings of the 29th ACM SIGKDD Conference on Knowledge Discovery and Data Mining, 2023.
[3] V Shejwalkar, A Houmansadr, "Manipulating the byzantine: Optimizing model poisoning attacks and defenses for federated learning, " In NDSS, 2021.

**Questions:**

1. Please show the performance of the proposed method when the server cannot obtain the training data and enough training data.
2. Please add some advanced defense methods in the experiments, e.g., [1] and [2].
3. The datasets used in this paper are too simple and limited to image classification tasks. Some real-world datasets, e.g., imagenet and cifar100, and other tasks should be considered, e.g., sentiment classification and multi-modal tasks.
4. The federated self-supervised learning framework should be considered in the experiments.
5. The convergence analysis is not meaningful, due to some impractical assumptions, e.g., Polyak-Łojasiewicz (PL) inequality.
6. The attacker setting in this paper is also simple. There are some advanced model poisoning attack methods, e.g., [3] and etc.

---

### Official Review · Reviewer_C2i6 · 2023-10-31

**Soundness:** 2 fair
**Presentation:** 2 fair
**Contribution:** 2 fair
**Rating:** 3
**Confidence:** 4

**Summary:**

This paper aims to defend against data poisoning attacks in federated learning FL. The authors aim to handle the trade-off between precision and robustness and circumvent the non-IID and strong-convexity assumption. In particular, they propose an optimal-transport-based defense called FLOT. FLOT optimizes the weighted Wasserstein distance to obtain weights for client updates. Experiments on benchmark datasets validate the effectiveness of the proposed FLOT.

**Strengths:**

* The discussed topic that defending against data poisoning attacks is interesting.
* The author provides the time complexity analysis of the proposed method.

**Weaknesses:**

1. The writing is confusing.
   1. Definition 3.1 is not a definition. I can't see any definition from it. It seems to be a proposition or a theorem about the vulnerability of FedAvg for me.
   2. Definition 4.1 is not clear.
      1. Eq (4) is confusing. According to the definition of $\mathbb{N}$, $\mathbb{R}$ and $\mathbb{X}$, $\mathbb{N}=\mathbb{R}\cup\mathbb{X}$. Then why there are additional $\nabla W_1$ and $\nabla W_n$ except for elements in $\mathbb{R}\cup\mathbb{X}$?
      2. How is Definition related to aggregation rule $\mathcal{A}$? $\mathcal{A}$ only appears in Eq.(4). But vector defined in Eq.(4)is not used.
   3. The operation \ in lines 241 and 243, is not defined.
   4. FLOT cost matrix in Algorithm 1 is not defined.
   5. Why the input of FLOT in Eq. (10) is different from previous ones?
2. The proposed method requires a validation dataset on the server, which violates privacy principle in FL.
3. It seems that the convergence result in Eq. (15) cannot guarantee that the global model converge to a good model rather than a bad model?

**Questions:**

1. The writing is confusing.
   1. Definition 3.1 is not a definition. I can't see any definition from it. It seems to be a proposition or a theorem about the vulnerability of FedAvg for me.
   2. Definition 4.1 is not clear.
      1. Eq (4) is confusing. According to the definition of $\mathbb{N}$, $\mathbb{R}$ and $\mathbb{X}$, $\mathbb{N}=\mathbb{R}\cup\mathbb{X}$. Then why there are additional $\nabla W_1$ and $\nabla W_n$ except for elements in $\mathbb{R}\cup\mathbb{X}$?
      2. How is Definition 4.1 related to aggregation rule $\mathcal{A}$? $\mathcal{A}$ only appears in Eq.(4). But vector defined in Eq.(4)is not used.
   3. The operation \ in lines 241 and 243, is not defined.
   4. FLOT cost matrix in Algorithm 1 is not defined.
   5. Why the input of FLOT in Eq. (10) is different from previous ones?
2. The proposed method requires a validation dataset on the server, which violates privacy principle in FL.
3. It seems that the convergence result in Eq. (15) cannot guarantee that the global model converges to a good model rather than a bad model.

---

### Official Review · Reviewer_b3Wa · 2023-11-05

**Soundness:** 3 good
**Presentation:** 3 good
**Contribution:** 3 good
**Rating:** 5
**Confidence:** 4

**Summary:**

The paper presents Federated Learning Optimal Transport (FLOT), a method for defending against black-box data poisoning attacks in federated learning. The method uses the Wasserstein barycentric technique to compute the global model and a loss function-based rejection for removing the effect from malicious clients. This requires the aggregator to have a trusted validation dataset representative of the global data distribution. The authors show the effectiveness of their defense compared to other methods in the research literature against some black-box data poisoning attacks in federated learning.

**Strengths:**

+ The use of the barycentric technique, borrowed from optimal transport, for computing a robust global model is an interesting approach for defending against some attacks in federated learning.
+ The authors provided a nice analysis of the computational complexity of the algorithm and the convergence analysis.
+ The paper is overall well presented, and the assumptions made are clear.

**Weaknesses:**

- In the presentation of FLOT the authors systematically compare with KRUM, which is a robust aggregation scheme that relies on a different set of assumptions. The comparison with other robust aggregation methods (e.g., FLTrust) that use similar assumptions is missing. It would be necessary to discuss how FLOT differentiate from these methods.
- The assumptions in the threat model are a bit restrictive: the authors just consider black-box data poisoning attacks and a strong aggregator that have a trusted validation set, representative of the global data distribution. It would be interesting to evaluate FLOT’s performance against more advanced model poisoning, adaptive, or more targeted attacks. In this sense, a more comprehensive evaluation is missing. Similarly, given that the method requires the aggregator to have a trusted validation dataset, a sensitivity analysis on the robustness of the method given the size of the dataset is also missing.

**Questions:**

- The comparison of the method with KRUM is not really fair, as KRUM relies on a different set of assumptions (e.g., it does not require having access to a separate validation dataset). In this sense, how does FLOT compare with other methods in the state of the art that also require access to a trusted validation dataset?
- In the experiments, it seems that FLTrust performs poorly in all the scenarios considered. This really contrasts the results provided by Cao et al. Could the authors explain this poor performance and the differences in the results with respect to Cao et al.’s paper?
- Having access to a separate validation dataset representative of the global training distribution is a strong assumption, which may not be possible for many applications of federated learning (e.g., in many cross-device applications) and, even if this is the case, gathering a large trusted dataset can be challenging. Could the authors provide a sensitivity analysis for assessing the robustness of the method as a function of the size of this trusted dataset?